# Spatial-temporal changes in runoff and terrestrial ecosystem water retention under 1.5°C and 2°C warming scenarios across China

Ran Zhai[1,2], Fulu Tao[1,2,3,*], Zhihui Xu[4]

[1] Key Laboratory of Land Surface Pattern and Simulation, Institute of Geographic Sciences and Natural Resources Research, Chinese Academy of Sciences, Beijing 100101, China
[2] College of Resources and Environment, University of Chinese Academy of Sciences, Beijing 100049, China
[3] Natural Resources Institute Finland (Luke), FI-00790 Helsinki, Finland
[4] Information Center of Yellow River Conservancy Commission, Zhengzhou 450004, China

*Correspondence to*: Fulu Tao (taofl@igsnrr.ac.cn)

**Abstract.** The Paris Agreement set a long-term temperature goal of holding the global average temperature increase to below 2.0°C above pre-industrial levels, pursuing efforts to limit this to 1.5°C, it is therefore important to understand the impacts of climate change under 1.5°C and 2.0°C warming scenarios for climate adaptation and mitigation. Here, climate scenarios from four Global Circulation Models (GCMs) for the baseline (2006-2015), 1.5°C and 2.0°C warming scenarios (2106-2115) were used to drive the validated Variable Infiltration Capacity (VIC) hydrological model to investigate the impacts of global warming on runoff and Terrestrial Ecosystem Water Retention (TEWR) across China at a spatial resolution of 0.5 degree. The trends in annual mean temperature, precipitation, runoff and TEWR were analysed at the grid and basin scale. Results showed that there were large uncertainties in climate scenarios from different GCMs, which led to large uncertainties in impact assessment. The differences among the four GCMs were larger than differences between the two warming scenarios. The interannual variability of runoff increased notably in areas where it was projected to increase, and the interannual variability increased notably from 1.5°C to 2.0°C warming scenario. By contrast, TEWR would remain relatively stable. Both low and high runoff would increase under the two warming scenarios in most areas across China, with high runoff increasing more. The risk of low and high runoff events would be higher under 2.0°C than 1.5°C warming scenario in term of both extent and intensity. Runoff was significantly positively correlated to precipitation, while increase in maximum temperature would generally cause runoff to decrease through increasing evapotranspiration. Likewise, precipitation also played a dominant role in affecting TEWR. Our findings on the spatiotemporal patterns of climate impacts and their shifts from 1.5°C to 2.0°C warming scenario are useful for water resource management under different warming scenarios.

## 1 Introduction

The global average surface temperature increased by 0.85°C from 1880 to 2012, and the beginning of the 21[st] century has been the warmest on record (IPCC, 2013). In 2015, the Paris Agreement set a long-term temperature goal of holding the global average temperature increase to below 2.0°C above pre-industrial levels, pursuing efforts to limit this to 1.5°C, because the

risks and impacts of climate change were thought to decrease significantly under global warming of 1.5℃ than 2.0℃ (Schleussner et al., 2016). This calls for spatial explicitly climate change impact assessment on multiple sectors under global warming of 1.5℃ and 2.0℃. Up to now, climate impact under 1.5℃ and 2.0℃ warming scenarios has been rarely assessed, but is urgently needed for climate adaptation and mitigation.

Global warming is likely to have major impacts on hydrological cycle (Huntington, 2006;Milliman et al., 2008;Arnell and Gosling, 2013), such as changing precipitation pattern and increasing risks of extreme hydrological events (Wang et al., 2012; Zhang et al., 2016). China is vulnerable to future climate change, the impacts of climate change on water resources in China have been of key concern (Piao et al., 2010;Leng et al., 2015). Hydrological models have been routinely used to investigate the impacts of climate change on water resources, driven by climate scenarios from Global Circulation Models (GCMs). Several
previous studies had assessed climate change impacts on water resources in some river basins over China (e.g, Chen et al., 2012; Li et al., 2013; Zhang et al., 2016). For example, with the Xin-anjiang model and HBV model in the Qingjiang Watershed, Chen et al. (2012) showed runoff would firstly decrease during 2011-2040 and then increase under A2 and B2 scenarios relative to the baseline period (1962-1990). Using the SIMHYD and GR4J rainfall-runoff models, driven by climate scenarios from 20 GCMs, mean runoff was projected to increase by most of the GCMs under a 1.0℃ increase in global average
surface air temperature across the Yarlung Tsangpo River basin (Li et al., 2013). Using the Soil and Water Assessment Tool (SWAT), Zhang et al. (2016) showed that future runoff would not change much under Representative Concentration Pathways 2.6 (RCP2.6) and 4.5 (RCP4.5) scenarios, but increased significantly under RCP8.5 scenario from three different GCMs (BCC-CSM1.1, CanESM2, and NorESM1-M) in the Xin River basin of China. However, climate change impact on water resources across the whole China has rarely been investigated. Using the Variable Infiltration Capacity (VIC) model, Wang et
al. (2012) showed the total amount of annual runoff over China would increase by approximately 3-10% by 2050 under A2, B2, and A1B emissions scenarios, with uneven distribution, relative to 1961-1990. Using the VIC model driven by climate scenarios from five GCMs under RCP8.5 emission scenario, Leng et al. (2015) showed that climate change could increase water-related risks across China in the 21[st] century because of projected decrease in runoff and increase in interannual variability. The changes in runoff under 1.5℃ and 2.0℃ warming scenarios across China have not been investigated yet.
Because of vast territory and large amount of population, it is important to understand the spatial explicitly changes in water resources under 1.5℃ and 2.0℃ warming scenarios across China.

      Terrestrial Ecosystem Water Retention (TEWR) is one of important processes affecting runoff yield. With the rapid growth of population and economy, ecosystem degradation and ecosystem services have increasingly become a hot topic. TEWR is one of important ecosystem services (Gong et al., 2017;Xu et al., 2017). Different methods have been used to
quantify TEWR. One of popular methods is based on terrestrial ecosystem water balance, the capacity of TEWR is the difference between the amount of precipitation and the sum of runoff and evapotranspiration (Ouyang et al., 2016;Xu et al., 2017). It is of great importance to evaluate TEWR service under changing climate for ecosystem and water resource management (Tao et al., 2003).

To our knowledge, this study is the first to investigate the changes in runoff and TEWR service across China under 1.5℃ and 2.0℃ warming scenarios, as well as the differences between the two warming scenarios. The objectives of this study are 1) to investigate the characteristics of expected changes in temperature and precipitation under 1.5℃ and 2.0℃ warming scenarios; 2) to investigate the changes in runoff, and TEWR across China under 1.5℃ and 2.0℃ warming scenarios at the grid scale and basin scale; 3) to evaluate the dominant factors for changes in runoff and TEWR under warming climate.

## 2 Materials and methods

### 2.1 Study domain

There are ten main basins in China (Fig. 1), including the Songhua River basin (SHR), Liao River basin (LR), Northwest River basins (NWR), Hai River basin (HR), and Yellow River basin (YR) in the northern China; the Yangtze River basin (YTR), Huai River basin (HuR), Southeast River basins (SER), Southwest River basins (SWR), and Pearl River basin (PR) in the southern China (Leng et al., 2015;Liu et al., 2017c). The temperature increases from north to south, and precipitation increases gradually from northwest to southeast (Xie et al., 2007). Mean annual runoff for China is around 284 mm based on synchronous runoff data for a 50-year period from 1956-2005 (Wang et al., 2012). However, water resource is unevenly distributed spatially and seasonally. In most areas, there are more than 70% of total runoff in the flood season from June to October (Wang et al., 2012). Water is more abundant in the southern China than the northern China (Piao et al., 2010).

### 2.2 Model description

A large-scale semi-distributed hydrological model, VIC, was applied in this study. We divided China into 0.5°×0.5° grids with three layers of soil in this study. The grids were between 18°N to 54°N from south to north, and between 73°E to 135.5°E from west to east. Only the grids with whole area located in the continental of China were investigated in this research. The soil and vegetation situation in each grid are considered in the model. Each grid cell is described by N+1 land cover tiles, it represents N different tiles of vegetation and bare soil. The total evapotranspiration includes canopy evaporation, vegetation transpiration and bare soil evaporation. The total runoff consists of surface runoff and base flow (Wang et al., 2012). The VIC model uses the variable infiltration curve to account for the spatial heterogeneity of runoff generation. It assumes that surface runoff for the upper two soil layers is generated by those areas for which precipitation exceeds the storage capacity of the soil. ARNO method is used to describe base flow, which only happens in the third layer of soil (Todini, 1996). A routing model is used to calculate runoff in each catchment after running VIC model (Lohmann et al., 1996). More details could be found in Liang et al. (1994, 1996), Liang and Xie (2001), Xie et al. (2003).

### 2.3 Data

Bias-corrected climate datasets for this study were from the project "Half a degree Additional warming, Prognosis and Projected Impacts" (HAPPI). It provides climate data to assess how the climate, especially extreme weather, might be

different from the current days in the world under 1.5°C and 2.0°C warmer than pre-industrial conditions (Mitchell et al., 2017). Large ensembles of simulations (>50 members) for three time periods have been produced after being bias corrected using the Inter-Sectoral Impact Model Intercomparison Project (ISIMIP2b) bias correction approach (Frieler et al., 2016), from four GCMs up to now, including ECHAM6-3-LR, MIROC5, NorESM1-HAPPI, and CAM4-2degree (Table 1) (http://portal.nersc.gov/c20c/data/ClimateAnalytics/). The first time period was from 2006 to 2015 which is the most recently observed 10-year, the second time period was from 2106 to 2115 under 1.5°C and 2.0°C warming scenarios, respectively. Each simulation within a time period was different from the others in its initial weather state (Mitchell et al., 2017). Table 1 shows the available ensemble members in each GCM under current period from 2006 to 2015, and 1.5°C and 2.0°C warming scenarios from 2106 to 2115. The input data of VIC includes daily precipitation, daily maximum temperature, daily minimum temperature and daily wind speed. And we analysed annual precipitation and annual mean temperature to show the differences between GCMs.

The obervation daily weather data, including daily time series of precipitation, maximum temperature, minimum temperature and wind speed, from 1961 to 1979 and from 2006-2015 was obtained from China Meteorological Administration (CMA) used for calibrate and validate the VIC model. The meteorological data were interpolated to each $0.5° \times 0.5°$ grid through linear interpolation weighted by the inverse squared distances between the meteorological stations and the center of each grid cells (Xie et al., 2007). 1 km land cover data was from the University of Maryland (http://glcfapp.glcf.umd.edu:8080/esdi/index.jsp). 1 km soil texture data (China Soil Map Based Harmonized World Soil Database (v1.1)), and 1 km Digital Elevation Model dataset were obtained from the Cold and Arid Regions Sciences Data Center at Lanzhou (http://westdc.westgis.ac.cn). These data were used to build the VIC model. The NASA Shuttle Radar Topographic Mission (SRTM) 90 m Digital Elevation Data (http://srtm.csi.cgiar.org/) was used to extract each catchment. Monthly runoff observation data, obtained from the hydrological year book of China and local water resources department, were used for calibrating and validating the VIC model. A detailed description was presented in Zhai and Tao (2017).

## 2.4 VIC model parameters calibration and validation

Monthly runoff data from 1961-1979 was used to calibrate and validate the VIC model (Wang et al., 2012;Liu et al., 2017c). Seven parameters in the VIC model needed to be calibrated because they were difficult to obtain, including the variable infiltration curve parameter (b), the maximum velocity of base flow (Dsmax), the fraction of Dsmax where non-linear base flow begins (Ds), the fraction of maximum soil moisture where non-linear base flow occurs (Ws), and the thickness of each soil moisture layer (di, i = 1,2,3). We divided 1961-1979 into three periods, including preheating period (1961-1962), calibration period (1963-1969), and validation period (1970-1979) in each catchment. The parameters calibrated in a catchment are further validated in other different catchment located in the same basin. The VIC model was run at daily time step and the results were aggregated to monthly time step at each catchment for calibrating and validating parameters. The relative error ( *BIAS* ; %) and the Nash-Sutcliffe efficiency coefficient ( *NSE* ) were used to calibrate and validate the parameters:

1) The *BIAS* (%) represents the error between simulated ($\bar{Q}_s$) and observed mean monthly runoff ($\bar{Q}_o$):

$$BIAS = (\bar{Q}_s - \bar{Q}_o)/\bar{Q}_o , \qquad (1)$$

2) The *NSE* (Nash and Sutcliffe, 1970) represents the matching degree between the simulated and observed runoff:

$$NSE = \frac{\sum(Q_{i,o} - \bar{Q}_o)^2 - \sum(Q_{i,o} - Q_{i,s})^2}{\sum(Q_{i,o} - \bar{Q}_o)^2} , \qquad (2)$$

where, $Q_{i,o}$ and $Q_{i,s}$ are the observed monthly runoff (mm) and the simulated monthly runoff (mm) at the month $i$, and $\bar{Q}_o$ is the mean observed monthly runoff (mm). A good simulation result will have *NSE* close to 1 and *BIAS* approach to 0.

## 2.5 Quantification of TEWR service

In this study, considering the input water and output water of a certain grid, we adopted the following equation to calculate the total amount of TEWR capacity (Ouyang et al., 2016;Xu et al., 2017).

$$W = P - ET - R , \qquad (3)$$

where, *W* represents TEWR (mm), *P* represents precipitation (mm), *ET* represents evapotranspiration (mm), and *R* represents runoff (mm).

## 2.6 Analysis

The climate scenarios from the GCMs of ECHAM6-3-LR, MIROC5, NorESM1-HAPPI, CAM4-2degree were input to drive the VIC hydrological model. Each GCM had output three climate change scenarios: baseline period from 2006-2015, 1.5℃ warming scenario from 2106-2115, and 2.0℃ warming scenario from 2106-2115. For each GCM of ECHAM6-3-LR, NorESM1-HAPPI, and CAM4-2degree, we had 200 simulations (10 years × 20 ensembles) for the baseline period, 1.5℃ and 2.0 ℃ warming scenarios, respectively. For the GCM of MIROC5, we had 100 simulations (10 years × 10 ensembles) for the baseline period, 1.5℃ and 2.0℃ warming scenarios, respectively. Changes in annual mean temperature and annual precipitation were calculated using each ensemble for future 10 years period (2106-2115) under 1.5℃ and 2.0 ℃ warming scenarios relative to the corresponding ensemble for the baseline period (2006-2015). Changes in annual mean and SD (standard deviation) as a measure of interannual variability were used to analyse the impacts of climate change on runoff and TEWR across China. We computed the changes in annual mean and SD of runoff and TEWR as the relative differences between the simulations using each ensemble for future 10 years period (2106-2115) under 1.5℃ and 2.0 ℃ warming scenarios relative to the simulations using the corresponding ensemble for the baseline period (2006-2015). For each warming scenario in each GCM, we adopted the median value of the changes among ensembles, which should be the most likely result avoiding abnormal value (Tao and Zhang, 2011). We also calculated the median value of annual mean temperature change, precipitation change, runoff change, TEWR change and the median value of SD change of runoff and

TEWR among all the 70 ensembles under the four GCMs of each grid. Then we calculated probability density functions of runoff change and TEWR change through the median value from all the 70 ensembles in every grid in the ten main basins across China under 1.5°C and 2.0°C warming scenarios (2106-2115) relative to the baseline period (2006-2015). The basin mean was calculated by averaging the values for the individual grid cells within the basin for each ensemble of a GCM.

Two annual runoff quantiles $Q_{10}$ (low runoff) and $Q_{90}$ (high runoff) were used to evaluate the risks of hydrological extremes. We used all ensemble simulations in the baseline period in 2006-2015, in 2106-2115 under 1.5°C warming scenario, and in 2106-2115 under 2.0°C warming scenario to evaluate the changes in low runoff and high runoff. Therefore, there were 700 years data (3 GCMs × 20 ensembles × 10 years + 1 GCM × 10 ensembles × 10 years = 700) for baseline period in 2006-2015, 1.5°C warming scenario in 2106-2115, and 2.0°C warming scenario in 2106-2115, respectively.

Pearson Correlation Coefficient $r$ was used to analyse the dominant factors affecting runoff and TEWR:

$$r = \frac{\sum_{i=1}^{n}(x_i - \overline{x})(y_i - \overline{y})}{\sqrt{\sum_{i=1}^{n}(x_i - \overline{x})^2 \sum_{i=1}^{n}(y_i - \overline{y})^2}} , \tag{4}$$

Where, n represents sample size, including four GCMs, two warming scenarios relative to the baseline in each GCM, 20 or 10 ensembles in each warming scenario, 10 years in each ensemble, so there are 1400 samples (3 GCMs × 2 warming scenarios ×20 ensembles ×10 years + 1 GCM ×2 warming scenarios × 10 ensembles × 10 years = 1400) in each grid. $x_i$ and $y_i$ are variable change values under 1.5°C and 2.0°C warming scenarios (2106-2115) relative to the baseline period (2006-2015) in each data set, and $\overline{x}$, $\overline{y}$ are mean change value of each variable in each grid.

# 3 Result

## 3.1 VIC model parameters calibration and validation

The VIC model was calibrated and validated in ten catchments located in different main basins in China (Fig. 1 and Fig. A1), and then the calibrated parameters were applied in all the grids located in the same basin. The *NSE* values of monthly runoff were above 0.70 in eight catchments in the calibration period, while the *NSE* values were above 0.70 in seven catchments in the validation period (Table 2). Except the Xiahui catchment, the *BIAS* values in all catchments were between -15% and 15%, which indicated the model simulated monthly runoff fairly well. Generally, the VIC model performed better in catchments located in the southern China where there were more precipitation and runoff compared with catchments located in the northern China. The *NSE* values for the Sanjiangkou, Xixian, Yangkou, Zhongaiqiao and Changle catchment in the southern China were all more than 0.75, and the *BIAS* values were between -10% and 10%. To make our parameter transplantation more convincing, we validated the calibrated parameters in ten other catchments different from those used for parameters calibration. Catchments areas range from 11280 km$^2$ to 730036 km$^2$. Generally,

results showed that the parameters calibrated in a catchment were also validated for other catchments located in the same basin, the *NSE* values for these ten catchments were all large than 0.65, and except the Bantai catchment, the *BIAS* value were all between -20% and 20% (Table 3).

## 3.2 Climate change across China under 1.5℃ and 2.0℃ warming scenarios

The median values of the changes in annual mean temperature and annual precipitation under the two warming scenarios for each GCM and all the four GCMs were shown in Fig. 2 and Fig. 3, respectively. Generally, the ECHAM6-3-LR and CAM4-2degree projected a relatively small increase in annual mean temperature (Figs. 2a,d,f,i). The MIROC5 projected a relatively large increase in annual mean temperature in comparison with other GCMs (Figs. 2b,g). As for annual precipitation, the ECHAM6-3-LR projected a decrease in precipitation over large areas across China under 1.5℃ warming scenario, and the decreasing trends reduced in the northeastern China and northwestern China, and increased in the Yellow River basin, Huai River basin, Yangtze River basin, and Southwest River basins under 2.0℃ warming scenario (Figs.3a,f). In contrast, the MIROC5, NorESM1-HAPPI and CAM4-2degree projected an increase in precipitation over large areas across China under both 1.5℃ and 2.0℃ warming scenarios. In particular, the MIROC5 projected the largest increase in precipitation by more than 20% in large areas in the southern China (Figs. 3b,g). Nearly all the four GCMs projected that precipitation decreased more (or increased less) in most areas located in the northwestern China than other areas in China (Fig. 3). In contrast, precipitation was projected to increase more (or decrease less) in the southeastern China (Fig. 3). There were large differences among the projections by the four different GCMs, suggesting a large uncertainty from climate change projection. Take all the 70 ensembles of the four GCMs as a whole, annual mean temperature was projected to increase more in the northern China and the middle and lower reaches of the Yangtze River basin than other areas (Figs. 2e,j). And annual precipitation was projected to increase more in the southeastern China under 1.5℃ warming scenario, and the increasing trend was projected to narrow down in the Yangtze River basin and extend to some areas located in the northern China under 2.0℃ warming scenario (Figs. 3e,j).

## 3.3 Changes in runoff across China under 1.5℃ and 2.0℃ warming scenarios

There were significant differences in projected change in runoff using the VIC model driven by the four different GCMs (Fig. 4). The projected runoff pattern was consistent with that of precipitation generally, suggesting precipitation change played a dominant role in runoff change. For example, under 1.5℃ and 2.0℃ warming scenarios by the ECHAM6-3-LR, runoff was projected to decrease in most areas across China (Figs. 4a,f) due to the projected decrease in precipitation (Figs. 3a,f). By contrast, using the climate scenarios by the MIROC5, runoff was projected to increase most (Figs. 4b,g) due to the projected increase in precipitation (Figs. 3b,g). In addition, increase in temperature would lead to increase in evapotranspiration generally (Fig. A2), which resulted in decrease in runoff. For example, under 1.5℃ warming scenario by the CAM4-2degree, precipitation would increase in most areas but the magnitude of increase was small, runoff was projected to decrease in large areas in the Hai River basin, Yellow River basin, Huai River basin, and the source regions of the Yellow River basin and

Yangtze River basin (Fig. 4d). Using the climate scenarios by the MIROC5 and NorESM1-HAPPI, runoff was projected to increase in most areas across China (Figs. 4b,c,g,h), suggesting the positive effects of precipitation increase should exceed the negative effects of temperature increase. For the median change across all the 70 ensembles in the four GCMs, runoff was projected to increase in large areas in China, especially in the Yellow River basin, Huai River basin, and Pearl River basin (Figs. 4e,j). By contrast, runoff was projected to decrease obviously in areas located in the Northwest River basins under 1.5℃ warming scenario and the source regions of the Yellow River basin and the Yangtze River basin under 2.0℃ warming scenario (Figs. 4e,j).

For each GCM, the median changes in SD among the ensembles under 1.5℃ and 2.0℃ warming scenarios were presented (Fig. 5). The SD was projected to increase notably in areas where the annual runoff increased notably, for all the four GCMs. Furthermore, the SD of the simulated runoff increased more under 2.0℃ warming scenario than that under 1.5℃ warming scenario generally (Figs. 5e,j), suggesting that interannual variation of runoff would increase with climate warming.

### 3.4 Changes in low runoff and high runoff across China under 1.5℃ and 2.0℃ warming scenarios

Both low runoff ($Q_{10}$) and high runoff ($Q_{90}$) were projected to increase in large areas across China, although decrease in some areas in the Northwest River basins and the source regions of the Yellow River basin and the Yangtze River basin (Fig. 6). High runoff was expected to increase more (or decrease less) than low runoff in most areas (Fig. 6). In some areas in the Northwest River basins, Songhua River basin and the source regions of the Yellow River basin and the Yangtze River basin, low runoff was projected to decrease under both 1.5℃ and 2.0℃ warming scenarios (Figs. 6a,b). The areas with low runoff decreasing were projected to enlarge under 2.0℃ warming scenario (Figs. 6a,b) around the source regions of the Yellow River basin and the Yangtze River basin, suggesting much more drought risks under 2.0℃ warming scenario than under 1.5℃ warming scenario. And the low runoff increased less under 2.0℃ warming scenario than 1.5℃ warming scenario in most grids in China (Figs. 6a,b). The areas with high runoff increasing were projected to enlarge under 2.0℃ warming scenario than 1.5℃ warming scenario (Figs. 6c,d), because the SD of annual precipitation was projected to increase more under 2.0℃ warming scenario than 1.5℃ warming scenario in most areas across China (Fig. A3). The intensity of high runoff was also expected to increase in most areas across China, especially in the Huai River basin (Figs. 6c,d), suggesting flood risks would increase under 2.0℃ warming scenario. In contrast, high runoff in some areas in the source regions of the Yellow River basin and the Yangtze River basin was expected to decrease, and decreased more under 2.0℃ warming scenario than 1.5℃ warming scenario (Figs. 6c,d), which caused by increasing evapotranspiration. Generally, high runoff was expected to increase more than low runoff in most areas across China, and the risks of high runoff and low runoff were expected to increase under 2.0℃ warming scenario than 1.5℃ warming scenario.

## 3.5 Changes in TEWR across China under 1.5°C and 2.0°C warming scenarios

Changes in TEWR were consistent with changes in precipitation and runoff. With the climate scenarios by ECHAM6-3-LR, TEWR was projected to decrease in large areas in China under both 1.5°C and 2.0°C warming scenarios (Figs. 7a,f), mainly due to the projected decrease in precipitation. In addition, precipitation was not the only factor for changes in TEWR. For example, precipitation was projected to increase in the source regions of the Yellow River basin and Yangtze River basin under warming scenarios by the MIROC5 (Figs. 3b,g), but TEWR there was projected to decrease (Figs. 7b,g) due to increasing evapotranspiration (Figs. A2b,g). Based on the median value of all 70 ensembles from the four GCMs, TEWR was projected to be more stable than runoff (Figs. 4e,j, Figs. 7e,j), projected changes for most grids would range from -5% to 5%. Compared with runoff, TEWR was projected to decrease under 2.0°C warming scenario than 1.5°C warming scenario relative to the baseline period (Figs. 4e,j, Figs. 7e,j).

As runoff, the SD of TEWR was projected to increase notably in areas where the TEWR increased notably for all the four GCMs generally (Fig. 8). However, the SD was projected to increase in some areas where TEWR decreased, such as the Liao River basin under 2.0°C warming scenario by the ECHAM6-3-LR (Fig. 7f, Fig. 8f). The changes in SD of TEWR were not as significant as that of runoff. The differences among the ten main basins were not as significant as runoff. As for the median SD value of the 70 ensembles, it was projected to increase more across China under 1.5°C (Fig. 8e) than 2.0°C warming scenario (Fig. 8j), suggesting the interannual variability of TEWR would be larger under 1.5°C warming scenario than 2.0°C warming scenario.

## 4 Discussion

### 4.1 Differences in climate variables and water resources between 1.5°C and 2.0°C warming scenarios by each GCM at the basin scale

To evaluate climate change and its potential impact on runoff and TEWR at the basin scale, the annual mean temperature change, annual precipitation change, annual runoff change, and annual TEWR change for all the 10 basins were summarized (Table 4 and Fig. 9). The uncertainties of all the 70 ensembles in each main basin of annual temperature change, annual precipitation change, annual runoff change and annual TEWR change were shown in Fig. A4. And the changes in low runoff ($Q_{10}$) and high runoff ($Q_{90}$) in each main basin in China from each GCM were shown in Fig. A5. At the basin scale, annual mean temperature increased more in the northern China (the median value ranged from 0.83°C to 0.92°C under 1.5°C warming scenario, and ranged from 1.55°C to 1.65°C under 2.0°C warming scenario) than the southern China (the median value ranged from 0.77°C to 0.86°C under 1.5°C warming scenario, and ranged from 1.41°C to 1.46°C under 2.0°C warming scenario), and the differences between the northern China and southern China enlarged under 2.0°C than 1.5°C warming scenario (Fig. 9a), while annual precipitation increased more in the southern China (the median value ranged from 3.86% to 8.58% under 1.5°C warming scenario, and ranged from 4.64% to 7.93% under 2.0°C warming scenario) than the northern China (the median value ranged from 1.90% to 5.24% under 1.5°C warming scenario, and ranged from 3.14% to 6.54%

under 2.0℃ warming scenario) (Fig. 9b). Generally, both runoff and TEWR would change consistently with precipitation (Figs. 9b,c,d). Annual mean temperature increased less under the climate scenarios by ECHAM6-3-LR than the other three GCMs in most basins. Annual precipitation was projected to decrease slightly in most basins under the climate scenarios by ECHAM6-3-LR. However, precipitation was projected to increase in all basins under the climate scenarios by the other three

GCMs (Fig. 9b). Runoff was projected to increase under the climate scenarios by the four GCMs, except some basins under the climate scenarios by ECHAM6-3-LR and CAM4-2degree (Fig. 9c). According to the median value across all ensembles in the four GCMs, runoff was projected to increase in all basins (the median value ranged from 3.61% to 13.86% under 1.5℃ warming scenario, and ranged from 4.20% to 17.89% under 2.0℃ warming scenario) but TEWR was projected to decrease or increase less than runoff (Figs. 9c,d) (the median value ranged from -0.45% to 6.71% under 1.5℃ warming

scenario, and ranged from -3.48% to 4.40% under 2.0℃ warming scenario). Our results showed that the differences of runoff and TEWR between GCMs were larger than those between warming scenarios by a certain GCM. This finding was supported by many other researches (Chen et al., 2011; Ouyang et al., 2015; Liu et al., 2017d; Zhang et al., 2017). In addition, we found that the changes in runoff and TEWR were more pronounced than those of precipitation. Precipitation changes ranged from -4.42% to 27.02%, however projected changes in runoff and TEWR would range from -20.12% to

84.02%, -18.57% to 34.84% of each GCM in every basin, respectively.

The variations of runoff under different GCMs and warming scenarios were larger in the Huai River basin than other basins (Fig. A4), and extremely large under the climate scenarios by MIROC5 (Fig. 9c), suggesting there could be larger uncertainty in the Huai River basin than other basins. The projected median annual mean precipitation and median annual mean runoff were, respectively, about 919 mm and 204 mm during 2006 to 2015 by the MIROC5 in the Huai River basin.

However, precipitation and temperature increase did not lead to a significant increase in evapotranspiration (Figs. A2b,g), more than 20% increase in precipitation would lead to a large percentage increase in runoff because the base value in the baseline period was small.

Probability density functions of median changes of runoff and TEWR across all ensembles in the four GCMs for all the grids in each basin were presented in Figs. A6 and A7, respectively. Runoff was projected to increase with higher probability

under 2.0℃ than 1.5℃ warming scenario in most basins across China (Fig. A6), because precipitation was projected to increase more under 2.0℃ than 1.5℃ warming scenario in most basins across China (Fig. 9b and Table 4). TEWR was projected to change less than runoff under both 1.5℃ and 2.0℃ warming scenarios (2106-2115) compared to the baseline period (2006-2015). TEWR change was projected to decrease with higher probability under 2.0 ℃ than 1.5℃ warming scenario compared with baseline period in most basins across China (Fig. A7). Probability density functions of changes of

low runoff and high runoff under 1.5℃ and 2.0℃ warming scenarios across all ensembles of the four GCMs for all the grids in each basin were presented in Figs. A8 and A9, respectively. Most grids under the two warming scenarios showed increasing low runoff and high runoff across China. High runoff was projected to increase more in most basins across China than low runoff under 2.0℃ warming scenario than 1.5℃ warming scenario (Figs. A8 and A9). And this, consistent with the

increase in SD of runoff under 2.0°C warming scenario (Fig. 5), may imply more flood and drought risks under 2.0°C warming scenario than 1.5°C warming scenario.

## 4.2 Major factors controlling changes in runoff, and TEWR

The VIC model has four input climate variables, including precipitation, maximum temperature, minimum temperature and wind speed, the Pearson Correlation Coefficient was calculated between the projected changes in annual runoff and the changes in climate variables, including annual precipitation (Fig. 10a), annual mean maximum temperature (Fig. 10b), annual mean minimum temperature (Fig. 10c) and annual mean wind speed (Fig. 10d). Only significant correlations were shown ($p<0.05$). Generally, the correlations between precipitation change and runoff change were much more significant than the other three variables in China (Fig. 10). This finding was supported by some previous studies (Dan et al., 2012;Wang et al., 2016;Huang et al., 2016;Liu et al., 2017c). The correlations were smaller in the Yellow River basin, Hai River basin, and Huai River basin than other basins, even less than 0.5 for some grids in the Northwest River basins and the source regions of the Yellow River basin (Fig. 10a). This may be caused by complex topography and land type, as well as the arid condition, which prevented the amount of water to form runoff (Liu et al., 2017a). There were significant negative correlations between annual runoff change and annual mean maximum temperature change in most areas across China. Increasing annual maximum temperature would lead to runoff decrease in most areas because of increasing evapotranspiration (Huang et al., 2016), especially in the southern China (Fig. 10b). The correlations between annual runoff change and annual mean minimum temperature change were not significant in nearly half of the studied grids in China, which was negative at most areas in the Hai River basin, Huai River basin, the source regions of the Yellow River basin and Yangtze River basin, and some areas in the Yellow River basin, Yangtze River basin, Pearl River basin, Southwest River basins and the Northwest River basins, and positive at other areas. Increase in annual mean minimum temperature would increase melting of snow or ice in the Tibetan Plateau and high latitude areas with cold weather regime, resulting in an increase in water supply to runoff (Liu et al., 2017a). So correlation between precipitation and runoff was smaller in some areas in the Tibetan Plateau. Since increase in minimum temperature was accompanied with increase in precipitation in most of the climate scenarios, there were positive correlations between minimum temperature change and runoff change at some areas. However, at other areas (e.g. Huai River basin, Hai River basin), increase in minimum temperature change would lead to decrease in runoff change, mainly caused by increasing evapotranspiration. There were significantly negative correlations between runoff change and wind speed change at most areas. Decrease in wind speed would lead to less evapotranspiration (She et al., 2017), consequently more runoff.

The TEWR was calculated through three variables, including precipitation, evapotranspiration, and runoff, we analysed the correlations between TEWR and the three variables. Like runoff, the correlation coefficients were also more significant between annual TEWR change and annual precipitation change (Fig. 11a), which suggested that increase in precipitation change would lead to increase in TEWR change, but not as significant as the correlations between precipitation change and runoff change (Fig. 10a). The Pearson Correlation Coefficients were smaller in the Southwest River basins than those in

other basins. The correlations were nearly the same but smaller between TEWR change and runoff change than those between TEWR change and precipitation change (Figs. 11a,c), because runoff change and precipitation change had a strong correlation (Fig. 10a). There were negative correlations between TEWR change and evapotranspiration change in the Songhua River basin, Liao River basin, Hai River basin, Huai River basin, Southeast River basin, Pearl River basin and the

source regions of the Yellow River basin and Yangtze River basin (Fig. 11b), increase in evapotranspiration change with increasing temperature change would lead to decrease in TEWR change. However, increase in evapotranspiration with increasing temperature would usually company with precipitation increasing, which led to a positive correlation between the evapotranspiration change and TEWR change at some basins.

## 4.3 Uncertainty analysis

Although previous research has not investigated the changes in runoff under 1.5°C and 2.0°C warming scenarios, runoff changes under RCP2.6 scenario were assessed. Since the end-of-century anthropogenic radiative forcing conditions used for 1.5°C warming scenario is the same as that for the RCP2.6 scenario, the warming level of the two scenarios are comparable (Mitchell et al., 2017). Our findings are supported by previous studies using the RCP2.6 scenario. For example, under the RCP2.6 scenario, the ensemble-averaged precipitation was expected to increase throughout China between 2015 and 2099

by 0.48% per decade relative to 1986-2005 (Liu et al., 2017b) among seven GCMs from the Coupled Model Intercomparison Project phase 5 (CMIP5). Precipitation would increase across most regions in China under RCP2.6 among 12 GCMs in 2070-2099 compared with 1960-1979, which is the main reason for runoff change in China (Liu et al., 2017d). Runoff was projected to decrease in the source regions of the Yellow River basin under RCP2.6 scenario among 12 GCMs (Zhang et al., 2017).

The HAPPI annual mean temperature and precipitation data were compared with the observed data, and the runoff and TEWR results driven by the HAPPI data and observation data were also compared in the baseline period (2006-2015) (Fig.12). Median values from all ensembles in every GCM and all the 70 ensembles of annual mean temperature, annual precipitation, annual runoff and annual TEWR were used to represent HAPPI data in each grid. The differences between projected and observed temperature were generally between -2°C to 2°C in around 65% areas in China, nevertheless the

grids located in the western China showed large differences between HAPPI data from observed data (Figs. 12a1-a5), because the number of meteorological stations was sparse in the western China than other areas. The differences of annual precipitation between the HAPPI data and the observed data ranged from -20% to 20% in more than 75% areas in China. The differences were smaller in the southern China than those in the northern China and western China (Figs. 12b1-b5). The differences between the projected runoff and TEWR using the HAPPI data and observed data ranged from -20% to 20% at

about 50% of the grids (Figs. 12c1-c5, d1-d5).

The ensemble numbers from the 70 ensembles of the four GCMs showing an increase trend in annual runoff and annual TEWR under 1.5°C and 2.0°C warming scenarios were presented in Figure 13. Runoff showed a consistent increase trend in most areas, especially in the southern China (Figs.13a,b), however a consistent decrease trend in the source regions in the

Yellow River basin and Yangtze River basin under 2.0℃ warming scenario (Fig.13b). Unlike runoff, TEWR was projected to change inconsistently in most areas (Figs.13c,d). The projected changes in runoff and TEWR had large uncertainties due to uncertainties in GCMs. It is hard to determine which GCM is better than others. Therefore, this study applied ensemble projections with multiple climate scenarios from multiple GCMs to provide a more comprehensive and robust results.

Human activities also have unavoidable impacts on water resources, more and more evidence showed that the influence of human activities on water resources is significantly enhanced (Jiang and Wang, 2016;Yuan et al., 2016;Liu et al., 2017c;Zhai and Tao, 2017). Human activities such as land use/cover changes and the increase in water withdrawal will affect runoff in the future, which is not taken into account in this study, because changing catchment characteristics may also generate larger uncertainties in simulation (Liu et al., 2017d). Although increase in runoff was projected in most areas across

China, the runoff may experience a decrease trend with the influence of human activities such as water withdrawal for life, industry and agriculture. Therefore, the impacts of human activities should be elaborated in further studies.

**5 Conclusions**

The validated VIC model was applied to simulate future hydrological processes driven by climate scenarios by four GCMs. In general, annual mean temperature increased more in the northern China than the southern China. On the contrary, annual

precipitation increased more in the southern China than the northern China. The projected changes in runoff and TEWR were consistent with the projected changes in precipitation, which were different for different GCMs under 1.5℃ and 2.0℃ warming scenarios. Annual runoff was projected to increase in most areas in China using climate scenarios by most of the four GCMs. The interannual variations of runoff was projected to increase notably in areas where annual runoff increased notably, leading to more extreme risks. The interannual variations would enlarge under 2.0℃ warming scenario compared with 1.5℃

warming scenario. Furthermore, the high runoff increased much more than the low runoff especially in the Huai River basin, and the risks of high runoff and low runoff would be enlarged under 2.0℃ warming scenario in comparison with 1.5℃ warming scenario. Annual TEWR was projected to change less than annual runoff. The interannual variations of TEWR were more stable than those of runoff. Multi-ensemble simulation results showed that precipitation change was the dominant factors for changes in runoff and TEWR. Maximum temperature had a negative correlation with runoff in most areas across

China because it would increase evapotranspiration. However, a large uncertainty is originated from different GCMs, so in this research we used a large ensemble simulation to provide a more comprehensive and convincing result. The impacts of human activities should be elaborated in further studies.

**Data availability**

All the data is available upon request. Please contact Fulu Tao at taofl@igsnrr.ac.cn.

**Acknowledgement**

This work was supported by the National Key Research and Development Program of China (No. 2017YFA0604703), and the National Science Foundation of China (Nos. 41571088, 41571493 and 31561143003). Funding support by Luke of

Finland through the strategic ClimSmartAgri project is also gratefully acknowledged. We acknowledge HAPPI core team and NERSC for data storage.

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

**Table 1.** The ensemble members in each GCM used in this study.

| GCM | Original institute | Original institute ID | Ensemble member | | |
|---|---|---|---|---|---|
| | | | 2006-2015 | 2106-2115 (+1.5℃) | 2106-2115 (+2℃) |
| ECHAM6-3-LR | Max Planck Institute for Meteorology, Hamburg, Germany & Deutsche Klimarechenzentrum, Hamburg, Germany | MPI-M | 20 | 20 | 20 |
| MIROC5 | Atmosphere and Ocean Research Institute, The University of Tokyo, Chiba, Japan; National Institute for Environmental Studies, Ibaraki, Japan; Japan Agency for Marine-Earth Science and Technology, Kanagawa, Japan | MIROC | 10 | 10 | 10 |
| NorESM1-HAPPI | NorESM Climate modeling Consortium | NCC | 20 | 20 | 20 |
| CAM4-2degree | ETH, Zurich, Switzerland | ETH | 20 | 20 | 20 |

**Table 2.** Information on the ten catchments (hydrological stations) used for calibrated and validated parameters and the performance of the VIC model for monthly runoff simulation in each catchment.

| Catchment | Basin | Area (km²) | Longitude | Latitude | Missing year | Calibration period (1963-1969) | | Validation period (1970-1979) | |
|---|---|---|---|---|---|---|---|---|---|
| | | | | | | NSE | BIAS (%) | NSE | BIAS (%) |
| Nianzishan | SHR | 13567 | 122°53′ | 47°29′ | | 0.66 | -12.3 | 0.72 | 1.7 |
| Liaoyang | LR | 8082 | 123°12′ | 41°16′ | | 0.82 | -9.5 | 0.61 | 4.9 |
| Yingluoxia | NWR | 10009 | 100°11′ | 38°48′ | | 0.88 | -1.4 | 0.87 | 4.7 |
| Xiahui | HR | 5340 | 117°10′ | 40°37′ | | 0.75 | 14.5 | 0.61 | -22.7 |
| Qinan | YR | 9805 | 105°40′ | 34°54′ | | 0.66 | 5.7 | 0.60 | -5.9 |
| Sanjiangkou | YTR | 15242 | 111°18′ | 29°35′ | | 0.84 | -9.9 | 0.91 | 4.1 |
| Xixian | HuR | 10190 | 114°44′ | 32°20′ | | 0.82 | -6.2 | 0.83 | 5.7 |
| Yangkou | SER | 12669 | 117°55′ | 26°48′ | 1962, 1966 | 0.91 | 2.5 | 0.90 | -3.1 |
| Zhongaiqiao | SWR | 3562 | 101°30′ | 23°21′ | 1964 | 0.78 | -5.3 | 0.80 | 5.0 |
| Changle | PR | 6645 | 109°25′ | 21°50′ | | 0.88 | -3.6 | 0.82 | 7.1 |

**Table 3.** Information on the additional ten catchments (hydrological stations) used for validating parameters calibrated in different catchments of the same basin.

| Catchment | Basin | Area (km$^2$) | Longitude | Latitude | Data period | Missing year | NSE | BIAS (%) |
|---|---|---|---|---|---|---|---|---|
| Lanxi | SHR | 27305 | 126°20′ | 46°15′ | 1981-1987 | | 0.67 | 10.1 |
| Guantai | HR | 17800 | 114°05′ | 36°20′ | 1963-1979 | | 0.71 | -8.5 |
| Yangjiaping | YR | 14124 | 107°44′ | 35°20′ | 1963-1979 | | 0.71 | -15.2 |
| Huayuankou | YR | 730036 | 113°39′ | 34°55′ | 1963-1979 | | 0.65 | -14.7 |
| Huangjiagang | YTR | 95217 | 111°31′ | 32°31′ | 1963-1972 | | 0.75 | 14.0 |
| Pingshan | YTR | 485099 | 104°10′ | 28°38′ | 1963-1979 | | 0.83 | -19.1 |
| Bantai | HuR | 11280 | 115°04′ | 32°43′ | 1963-1979 | | 0.67 | 34.7 |
| Xuren | SER | 13560 | 120°19′ | 28°09′ | 1964-1979 | | 0.81 | -16.1 |
| Changdu | SWR | 48448 | 97°11′ | 31°11′ | 1963-1979 | 1964, 1969, 1971, 1972 | 0.81 | -2.3 |
| Wuzhou | PR | 327006 | 111°20′ | 23°28′ | 1963-1979 | | 0.90 | 2.9 |

**Table 4.** Median values of changes in annual mean temperature (℃), annual precipitation (%), annual runoff (%) and annual TEWR (%) in all corresponding ensembles of the four GCMs over 2106-2115 under 1.5℃ and 2.0℃ warming scenarios relative to baseline period 2006-2015, respectively, in the ten main basins across China.

| | ΔT (℃) | | ΔP (%) | | ΔR (%) | | ΔTEWR (%) | |
|---|---|---|---|---|---|---|---|---|
| | +1.5℃ | +2.0℃ | +1.5℃ | +2.0℃ | +1.5℃ | +2.0℃ | +1.5℃ | +2.0℃ |
| SHR | 0.83 | 1.65 | 3.50 | 4.82 | 5.34 | 6.20 | 2.11 | -1.49 |
| LR | 0.92 | 1.62 | 5.24 | 6.54 | 6.86 | 9.07 | 1.76 | 0.65 |
| NWR | 0.87 | 1.63 | 1.90 | 3.14 | 3.61 | 4.20 | 0.52 | -0.17 |
| HR | 0.85 | 1.55 | 3.22 | 4.44 | 7.93 | 4.75 | 3.00 | 4.40 |
| YR | 0.88 | 1.57 | 4.06 | 5.72 | 8.72 | 9.40 | 2.84 | -0.67 |
| YTR | 0.86 | 1.46 | 5.05 | 4.64 | 7.07 | 4.33 | -0.45 | -3.48 |
| HuR | 0.80 | 1.44 | 5.94 | 6.68 | 12.11 | 17.89 | 6.71 | 4.01 |
| SER | 0.84 | 1.42 | 8.58 | 7.93 | 11.04 | 8.77 | 1.76 | -1.60 |
| SWR | 0.77 | 1.43 | 3.86 | 4.80 | 4.94 | 5.18 | -0.36 | -0.82 |
| PR | 0.84 | 1.41 | 6.71 | 6.26 | 13.86 | 11.17 | 2.79 | 0.37 |

ΔT, ΔP, ΔR, ΔTEWR represent annual mean temperature change, annual precipitation change, annual runoff change and annual TEWR change.

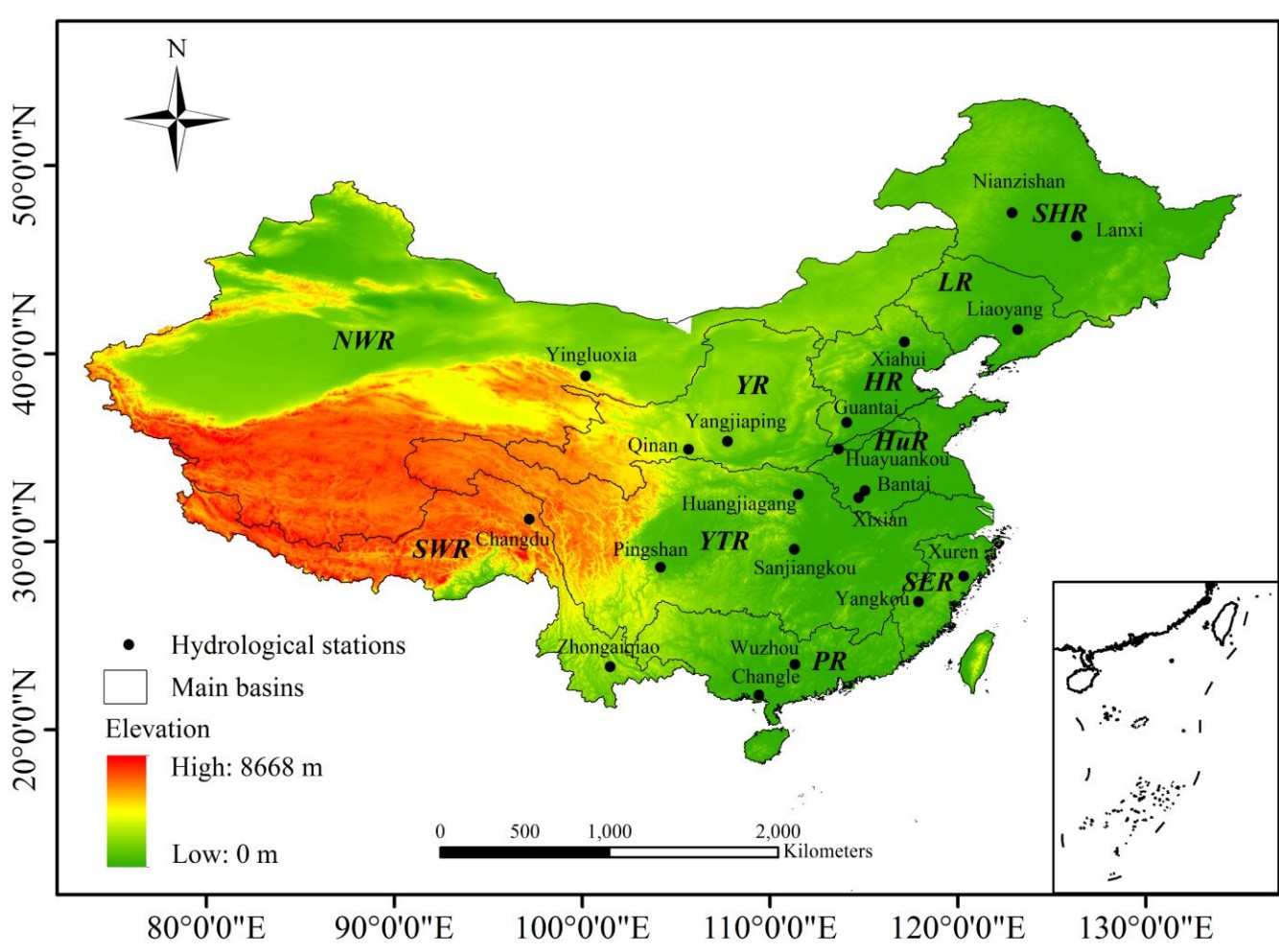

**Figure 1: The ten main basins in China, as well as the twenty catchments (hydrological stations) used for calibrated and validated the VIC hydrological model.**

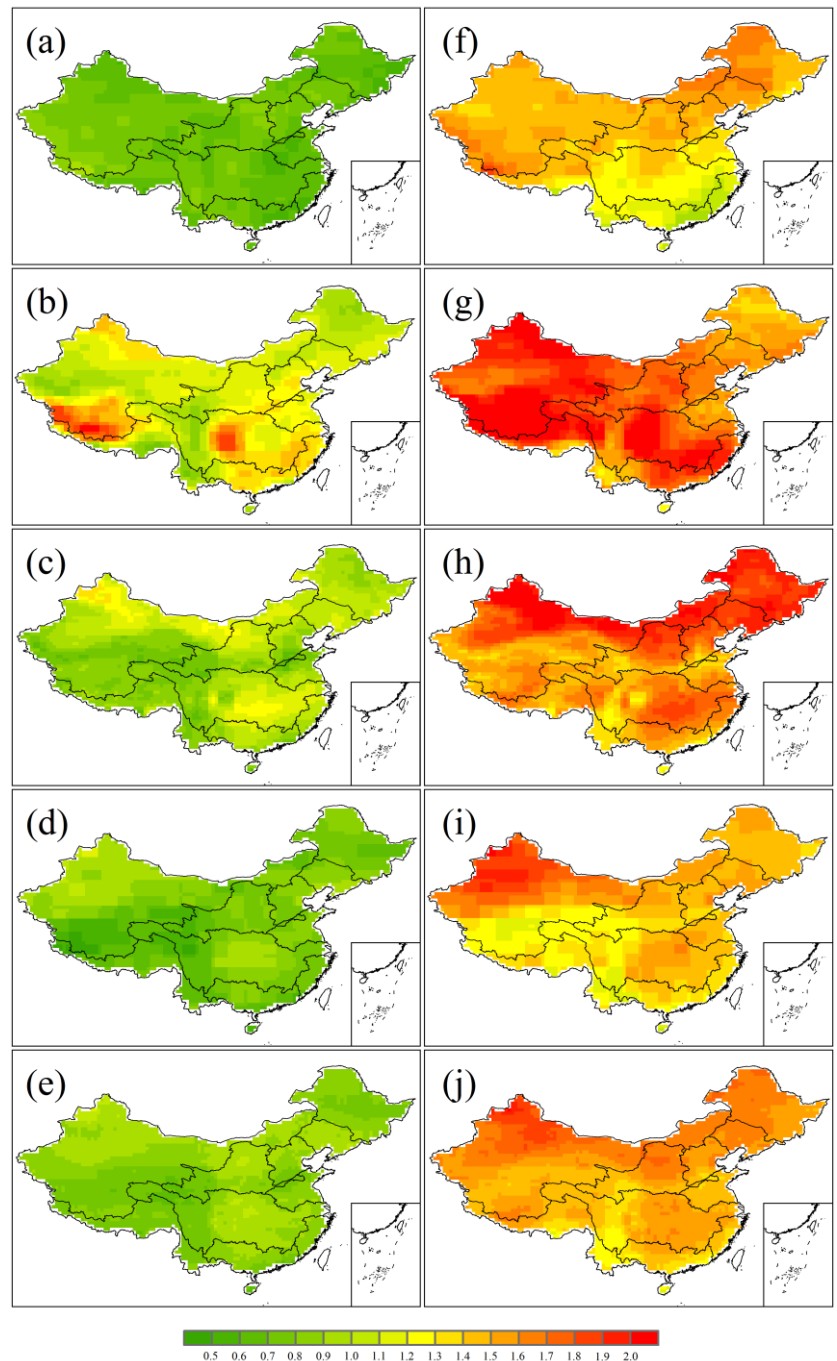

**Figure 2: Median values of the projected changes in annual mean temperature (°C) in China under the 1.5°C (a, b, c, d, e) and 2.0°C (f, g, h, i, j) warming scenarios by the ECHAM6-3-LR (a, f), MIROC5 (b, g), NorESM1-HAPPI (c, h), CAM4-2degree (d, i), all the four GCMs (e, j), relative to 2006-2015.**

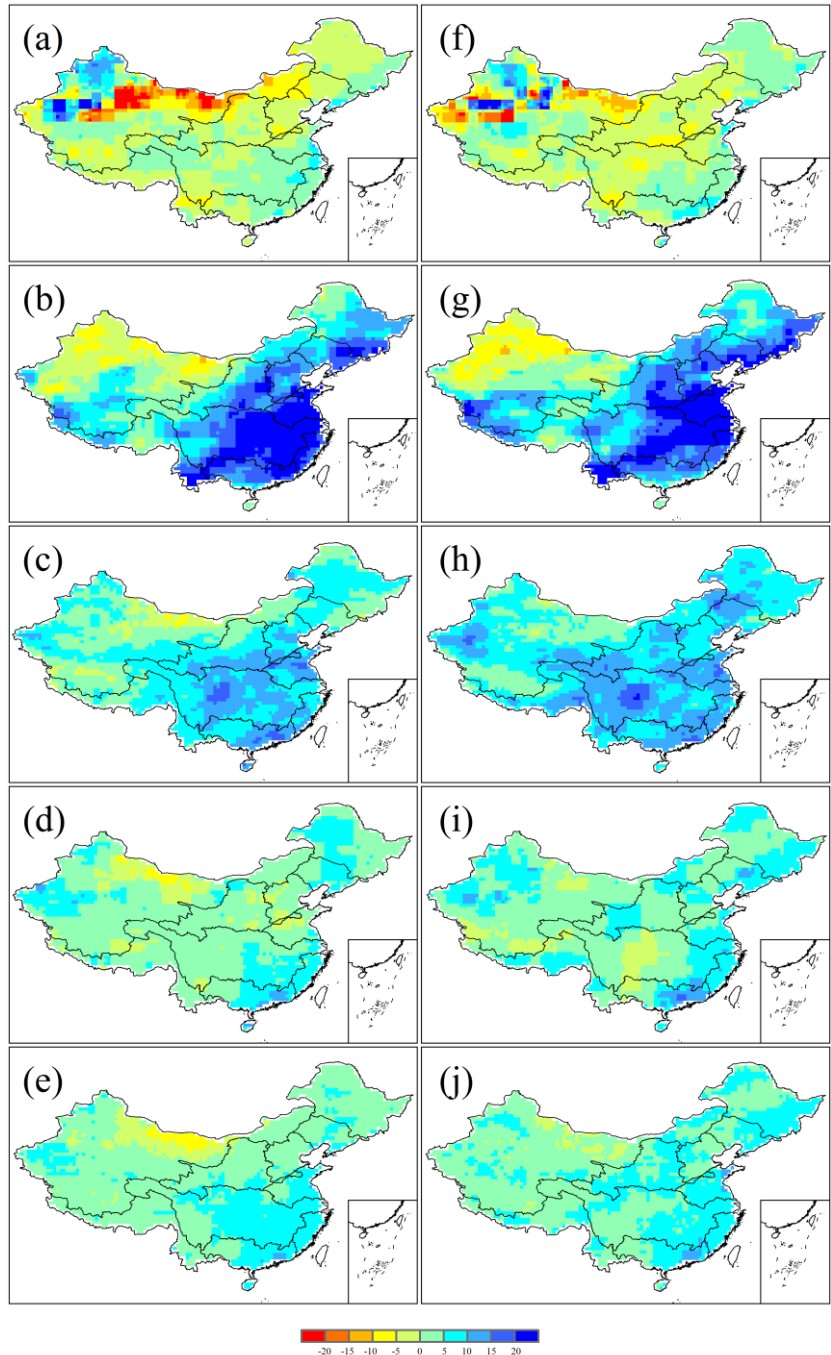

**Figure 3: Median values of the projected changes in annual precipitation (%) in China under the 1.5°C (a, b, c, d, e) and 2.0°C (f, g, h, i, j) warming scenarios by the ECHAM6-3-LR (a, f), MIROC5 (b, g), NorESM1-HAPPI (c, h), CAM4-2degree (d, i), all the four GCMs (e, j), relative to 2006-2015.**

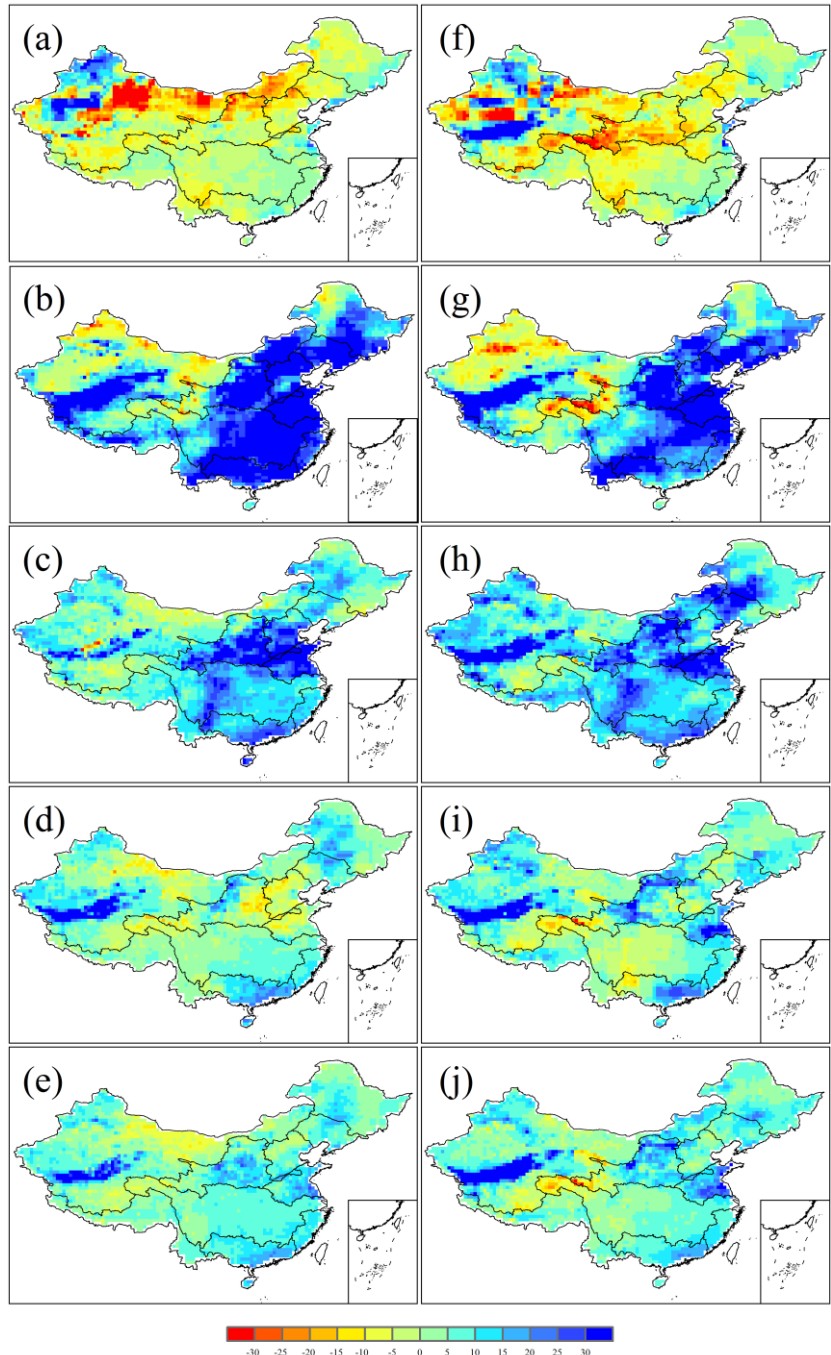

**Figure 4: Median values of the projected changes in annual runoff (%) in China under the 1.5°C (a, b, c, d, e) and 2.0°C (f, g, h, i, j) warming scenarios by the ECHAM6-3-LR (a, f), MIROC5 (b, g), NorESM1-HAPPI (c, h), CAM4-2degree (d, i), all the four GCMs (e, j), relative to 2006-2015.**

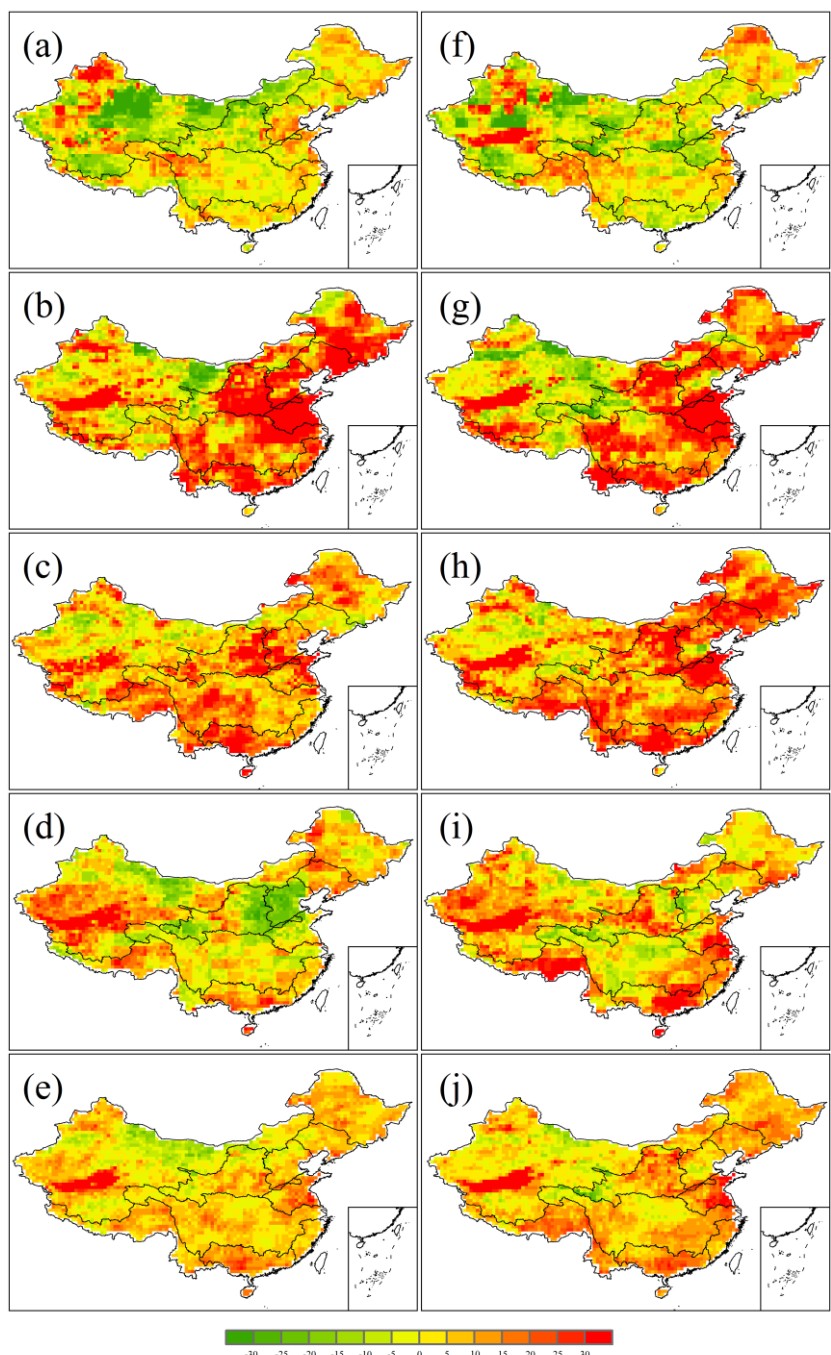

**Figure 5: Median values of the changes in SD of the simulated annual runoff (%) in China under the 1.5°C (a, b, c, d, e) and 2.0°C (f, g, h, i, j) warming scenarios by the ECHAM6-3-LR (a, f), MIROC5 (b, g), NorESM1-HAPPI (c, h), CAM4-2degree (d, i), all the four GCMs (e, j), relative to 2006-2015.**

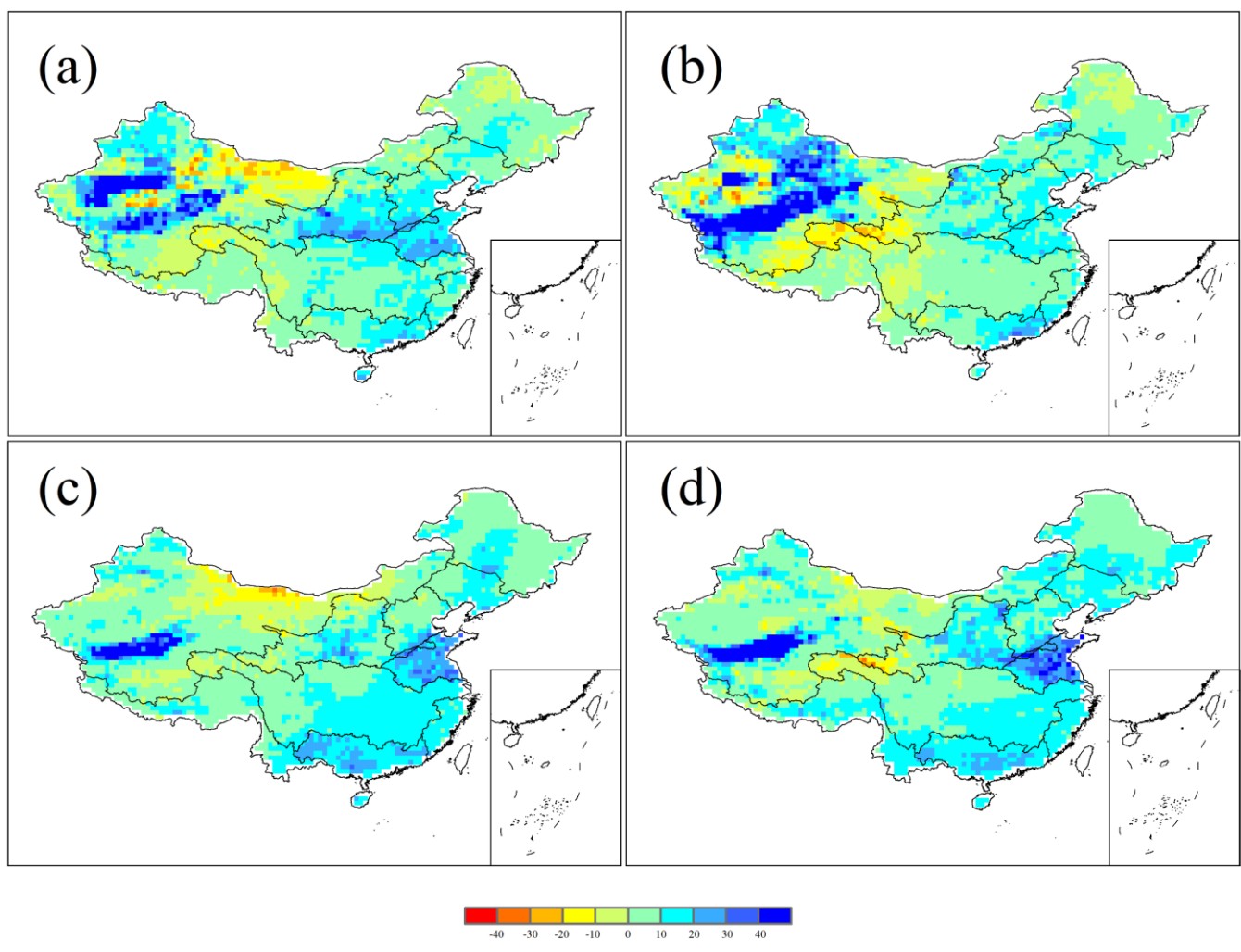

**Figure 6: Projected changes in low runoff (Q$_{10}$) and high runoff (Q$_{90}$) (%) in China under 1.5°C (a, c) and 2.0°C (b, d) warming scenarios in 2106-2115, respectively, relative to 2006-2015.**

.

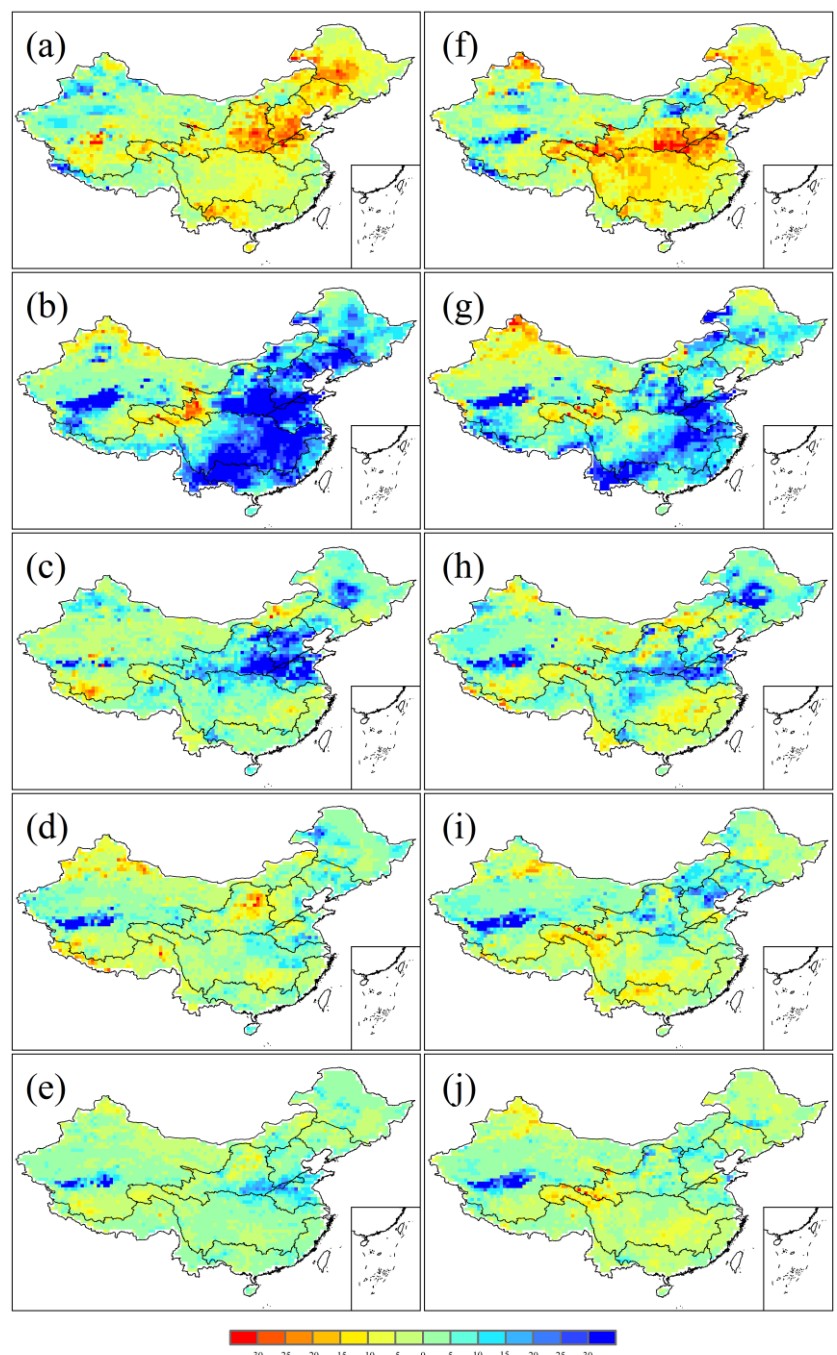

**Figure 7: Median values of the changes in annual TEWR (%) in China under 1.5°C (a, b, c, d, e) and 2.0°C (f, g, h, i, j) warming scenarios from the ECHAM6-3-LR (a, f), MIROC5 (b, g), NorESM1-HAPPI (c, h), CAM4-2degree (d, i), all the four GCMs (e, j), relative to 2006-2015.**

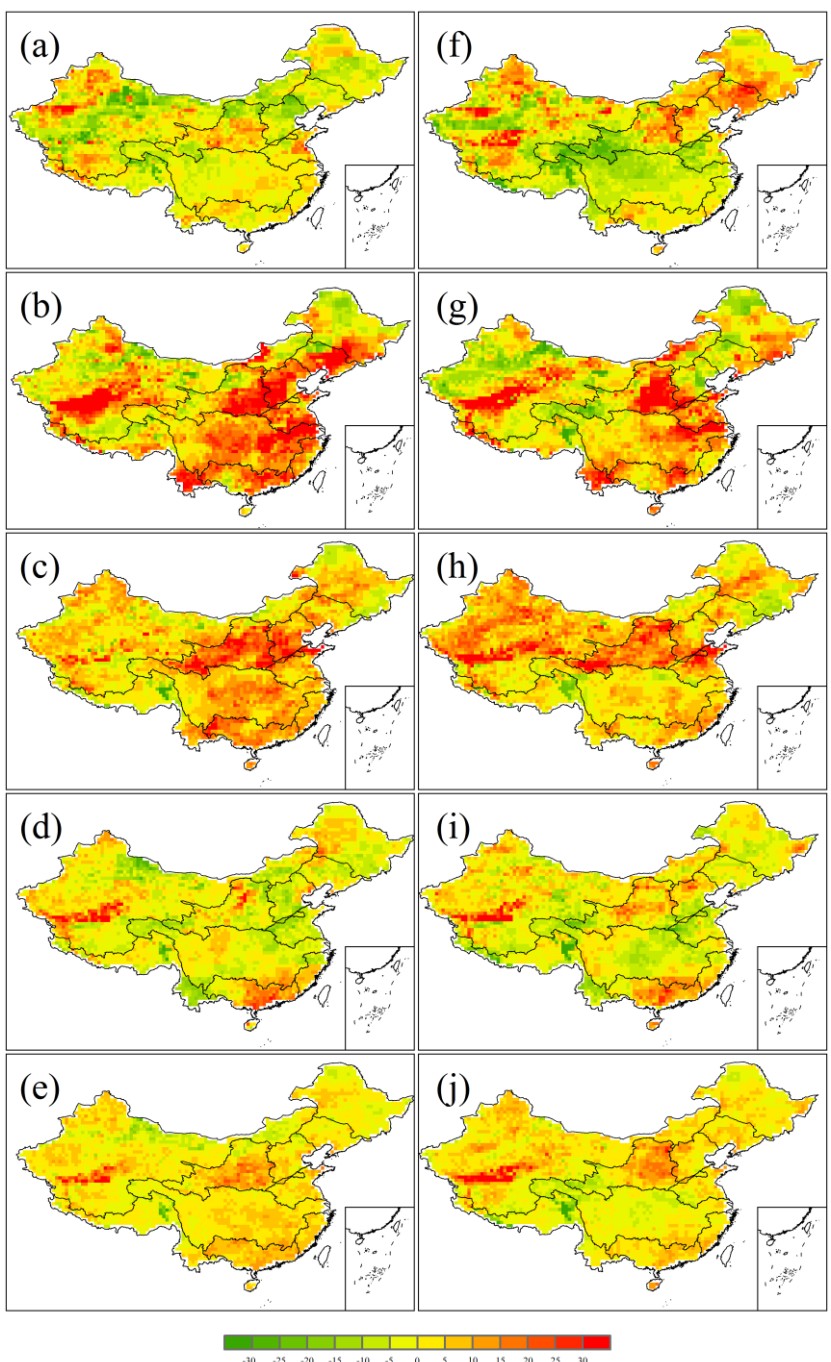

**Figure 8: Median values of the changes in SD of TEWR (%) in China under 1.5°C (a, b, c, d, e) and 2.0°C (f, g, h, i, j) warming scenarios from the ECHAM6-3-LR (a, f), MIROC5 (b, g), NorESM1-HAPPI (c, h), CAM4-2degree (d, i), all the four GCMs (e, j) , relative to 2006-2015.**

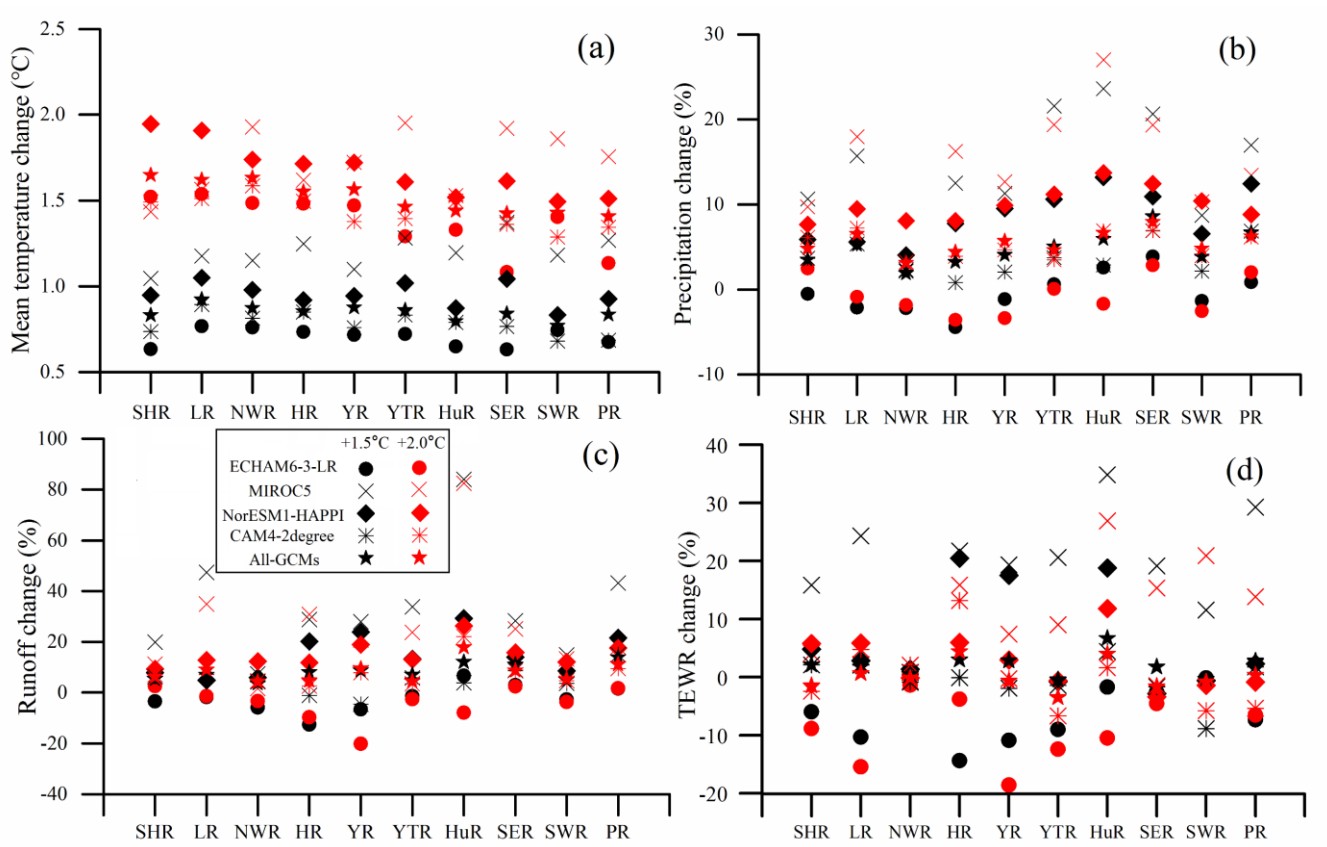

**Figure 9: Median values of changes in annual mean temperature (°C) (a), annual precipitation (%) (b), annual runoff (%) (c) and annual TEWR (%) (d) in all corresponding ensembles of the four GCMs during 2106-2115 under 1.5°C and 2.0°C warming scenarios relative to the baseline period 2006- 2015, respectively, in the ten main basins across China.**

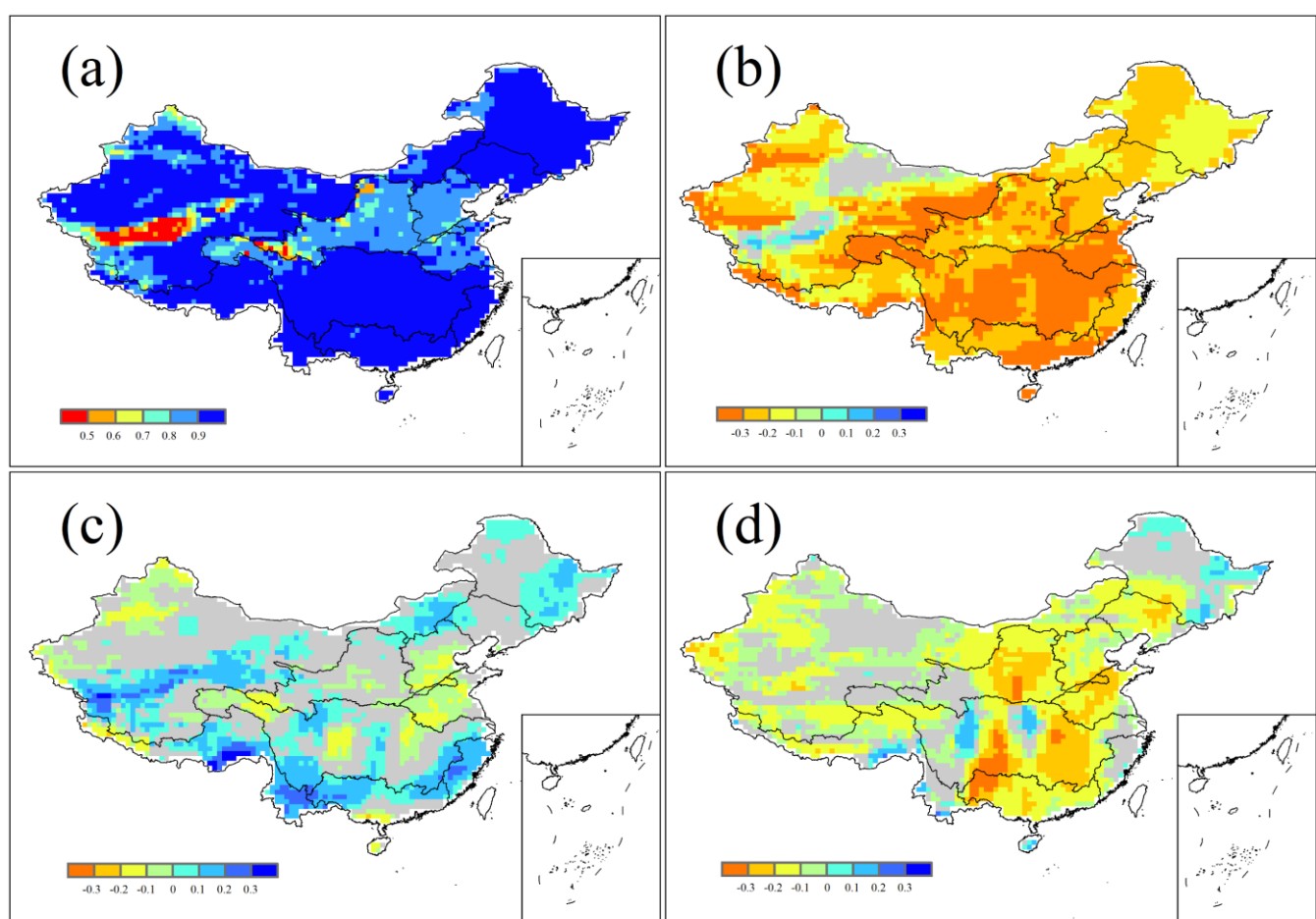

**Figure 10: Spatial patterns of the Pearson Correlation Coefficient (r) between data series of runoff changes and four key impact factors changes (a: runoff and precipitation, b: runoff and annual maximum temperature, c: runoff and annual minimum temperature, d: runoff and wind speed) under 1.5°C and 2.0°C warming scenarios during 2106-2115, relative to the baseline period 2006-2015. Only grids with significant correlation (p<0.05) were shown.**

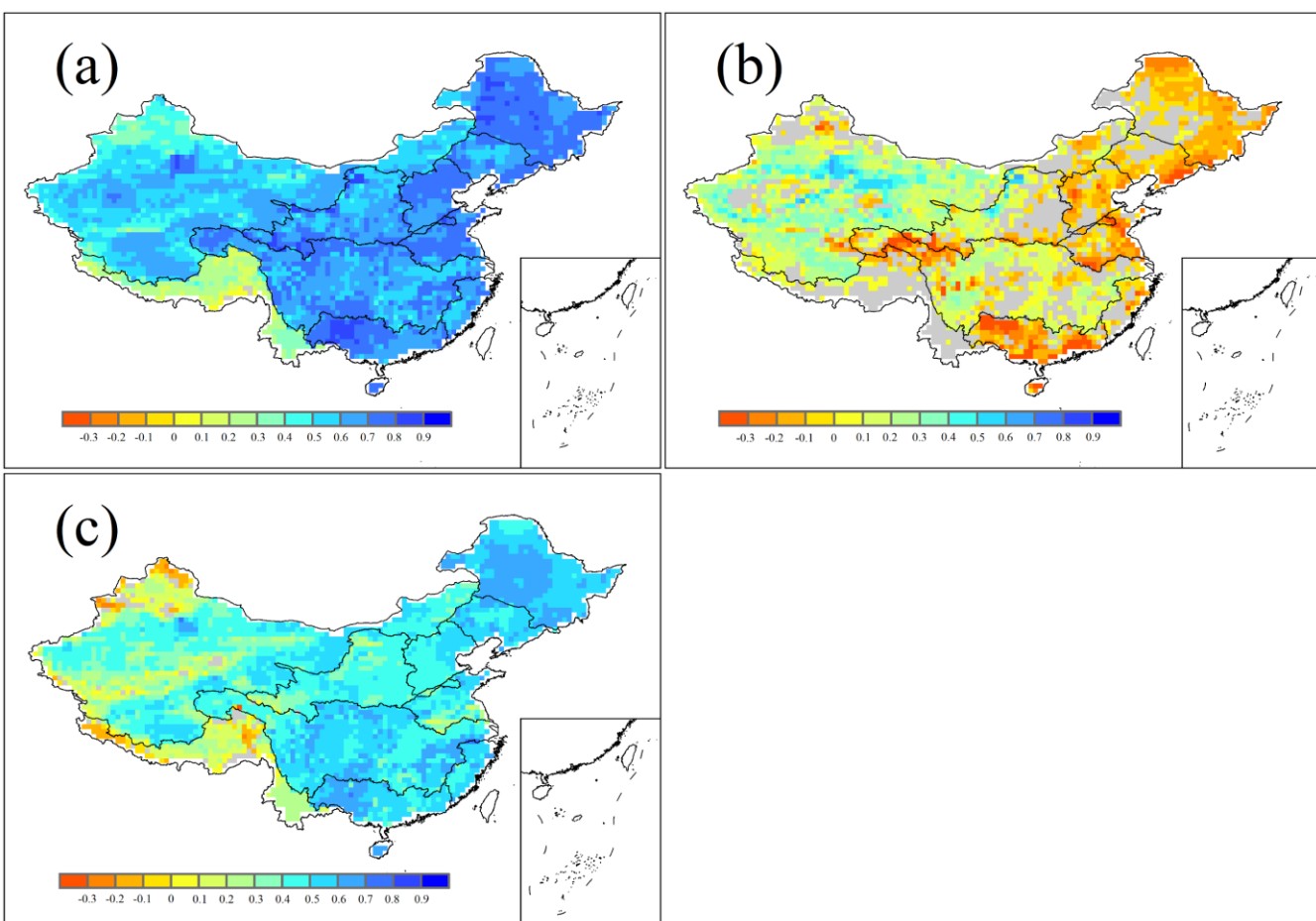

**Figure 11: Spatial patterns of the Pearson Correlation Coefficient (r) between data series of TEWR changes and three key impact factors changes (a: TEWR and precipitation; b: TEWR and evapotranspiration; c: TEWR and runoff) under 1.5°C and 2.0°C warming scenarios during 2106-2115, relative to the baseline period 2006-2015. Only grids with significant correlation (p<0.05) were shown.**

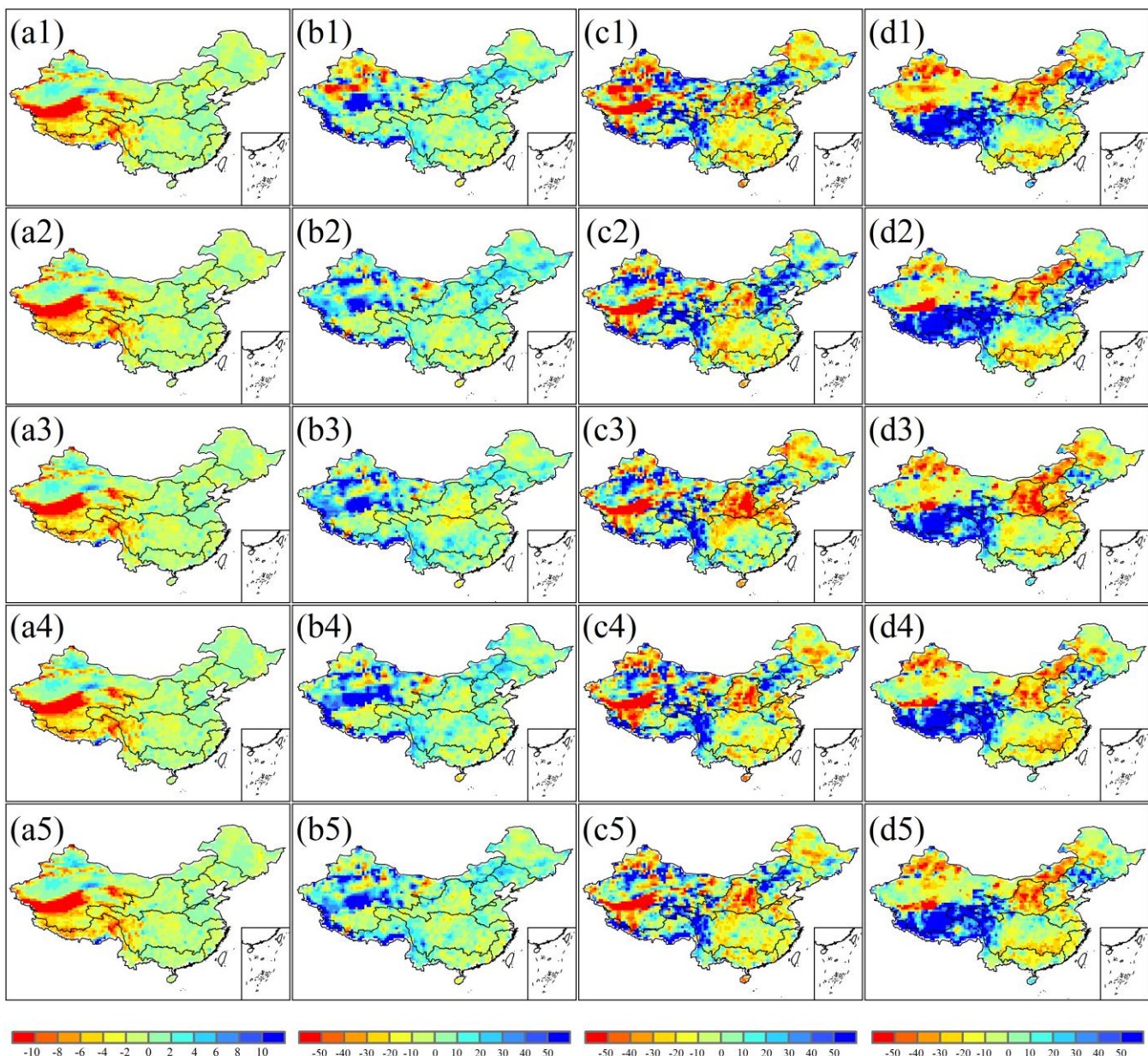

**Figure 12: Differences between the HAPPI data and the observed data in terms of annual temperature (°C) (a1-a5), annual precipitation (%) (b1-b5), as well as the differences between the projected annual runoff (%) (c1-c5) and annual TEWR (%) (d1-d5) by the HAPPI data and by the observed data, for 2006-2015, based on the ECHAM6-3-LR (a1, b1, c1, d1), MIROC5 (a2, b2, c2, d2), NorESM1-HAPPI (a3, b3, c3, d3), CAM4-2degree (a4, b4, c4, d4), and all the four GCMs (a5, b5, c5, d5), respectively.**

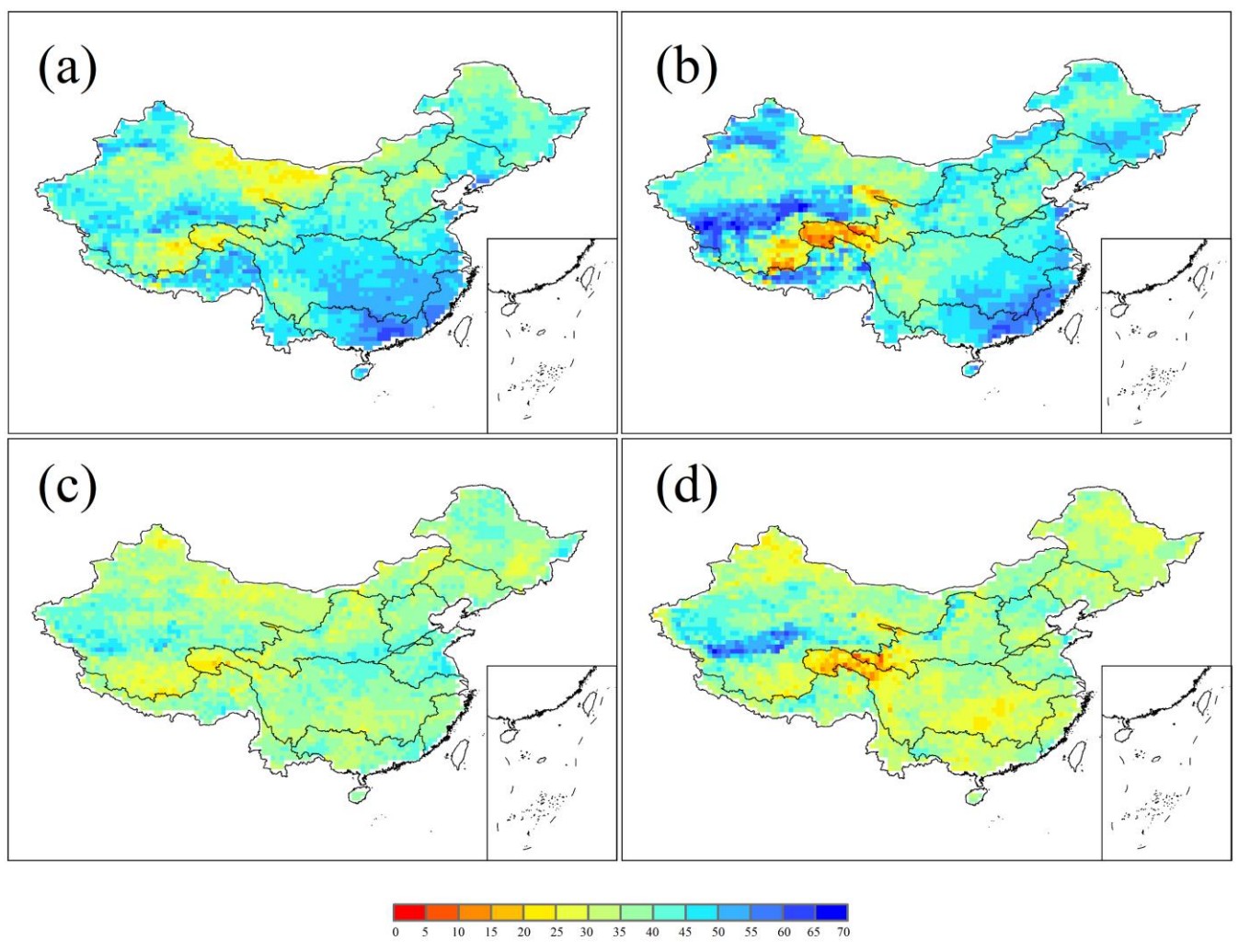

**Figure 13: Ensemble numbers out of 70 ensembles showing an increase in runoff (a, b) and TEWR (c, d) change under 1.5°C (a, c) and 2.0°C warming scenarios (b, d).**

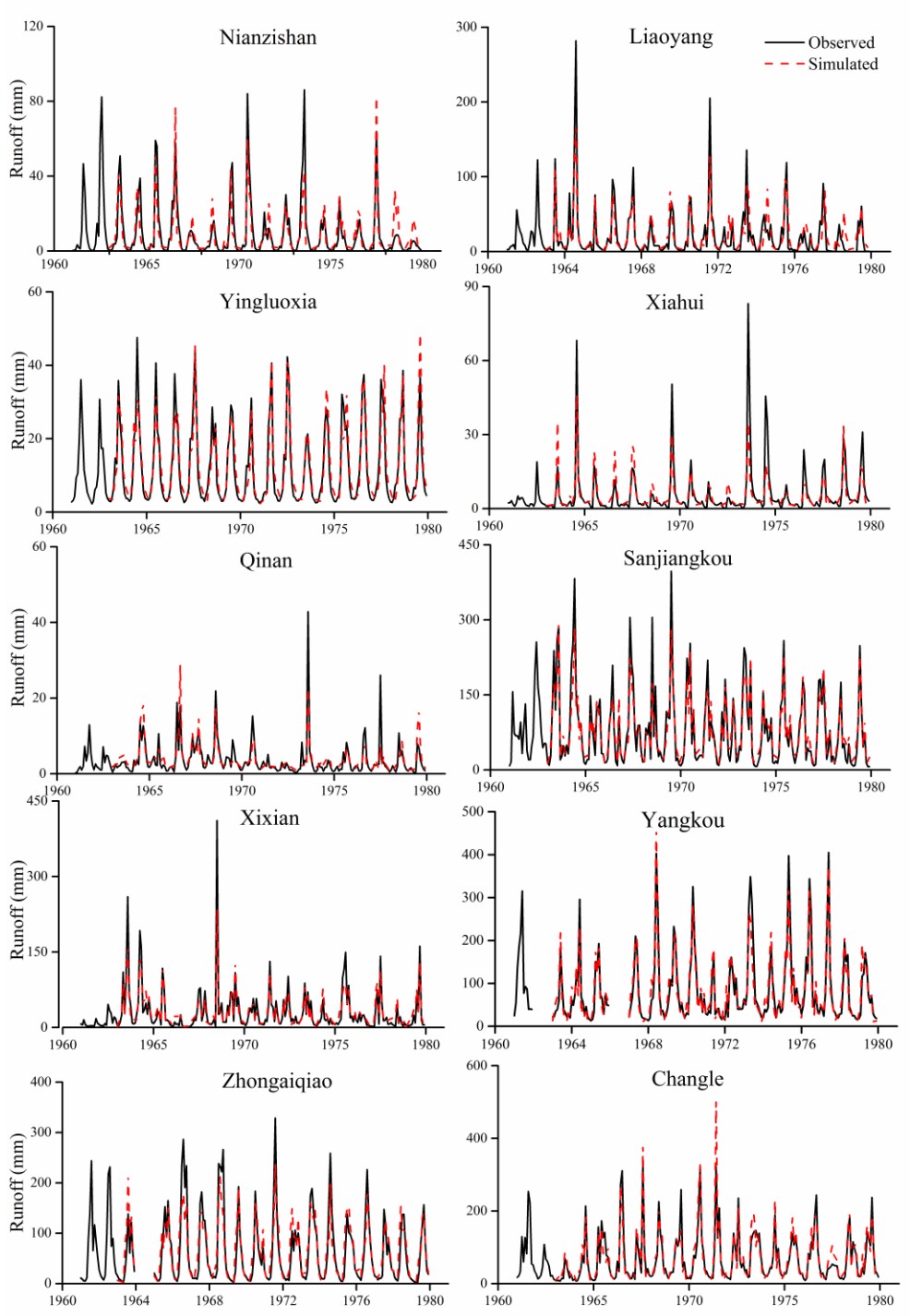

**Figure A1: Observed and simulated monthly runoff during the preheating period (1961–1962), calibration period (1963–1969) and validation period (1970–1979).**

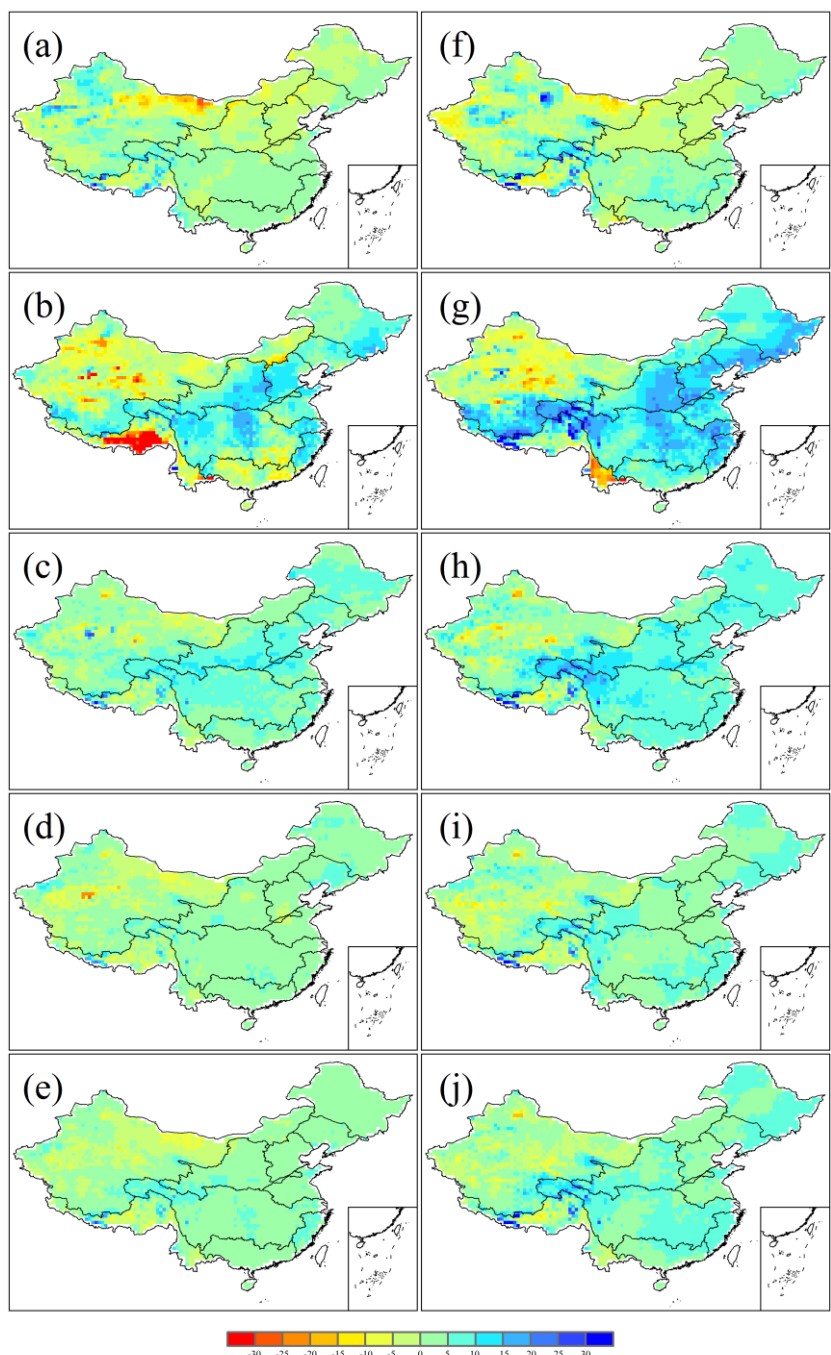

**Figure A2: Median values of the projected changes in annual evapotranspiration (%) in China under the 1.5°C (a, b, c, d, e) and 2.0°C (f, g, h, i, j) warming scenarios by the ECHAM6-3-LR (a, f), MIROC5 (b, g), NorESM1-HAPPI (c, h), CAM4-2degree (d, i), all the four GCMs (e, j), relative to 2006-2015.**

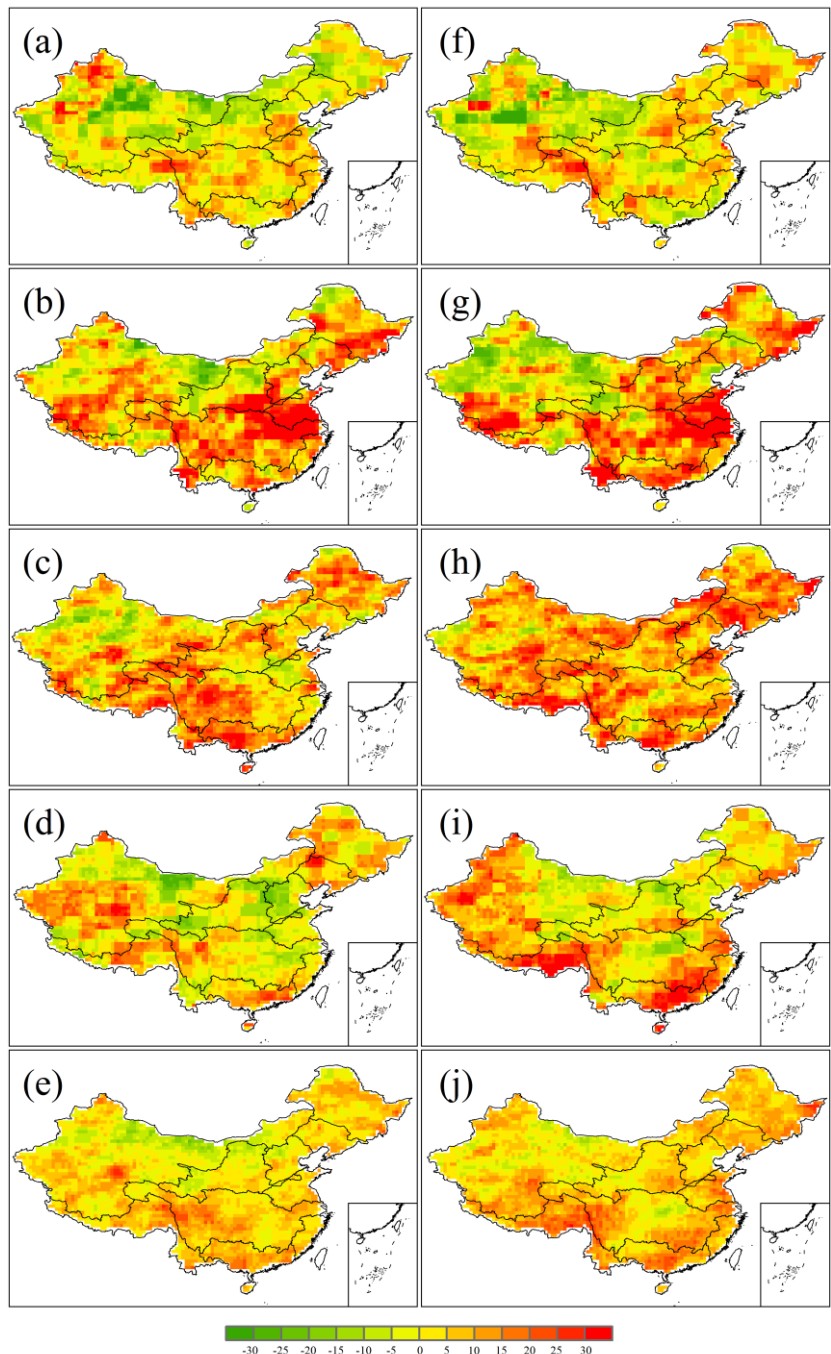

**Figure A3: Median values of the changes in SD of the annual precipitation (%) in China under the 1.5°C (a, b, c, d, e) and 2.0°C (f, g, h, i, j) warming scenarios by the ECHAM6-3-LR (a, f), MIROC5 (b, g), NorESM1-HAPPI (c, h), CAM4-2degree (d, i), all the four GCMs (e, j), relative to 2006-2015.**

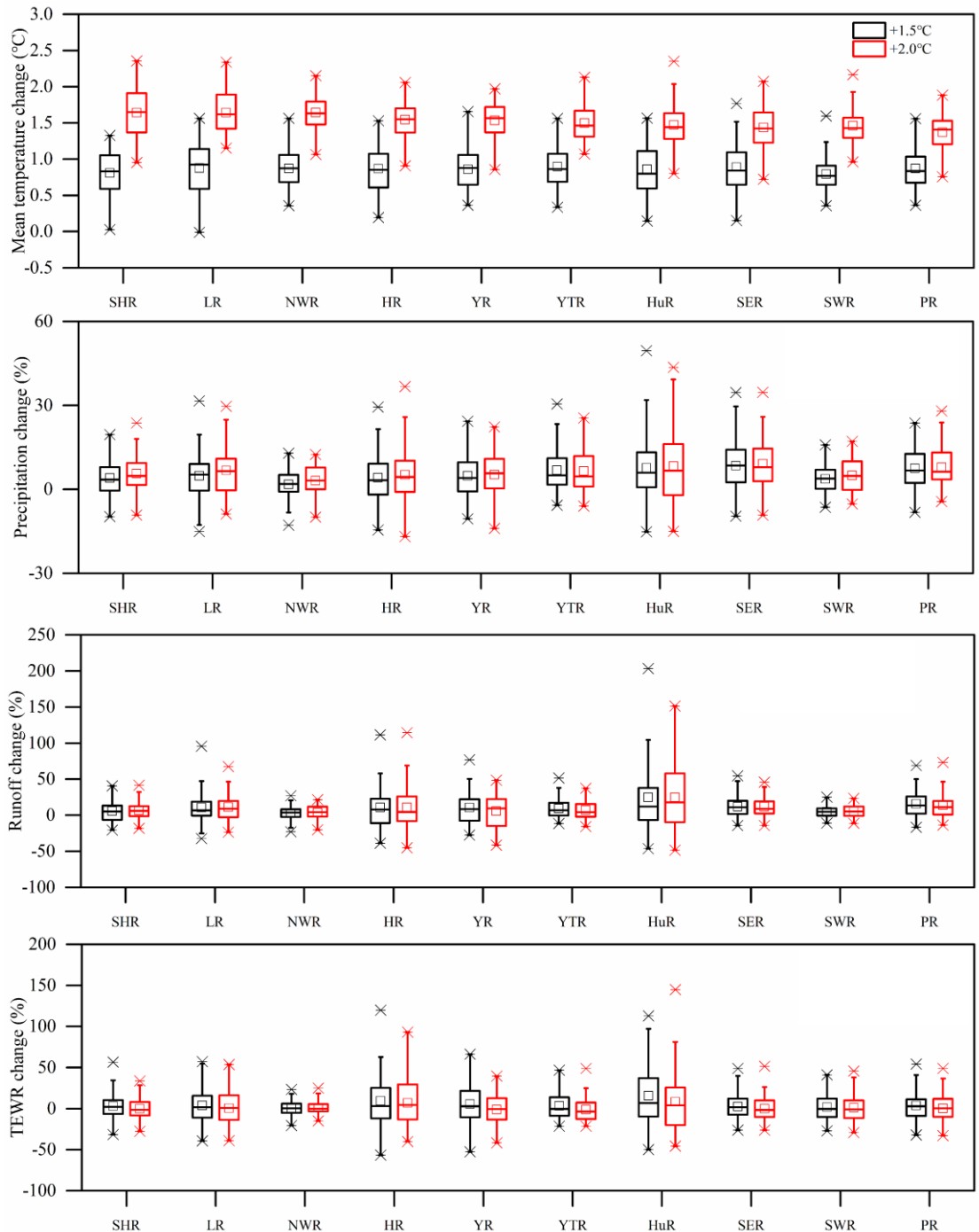

**Figure A4:** Box-and-whisker plots of annual mean temperature change (°C), annual precipitation change (%), annual runoff change (%), annual TEWR change (%) from all the 70 ensembles under 1.5°C and 2.0°C warming scenarios in the ten main basins across China, relative to 2006-2015.

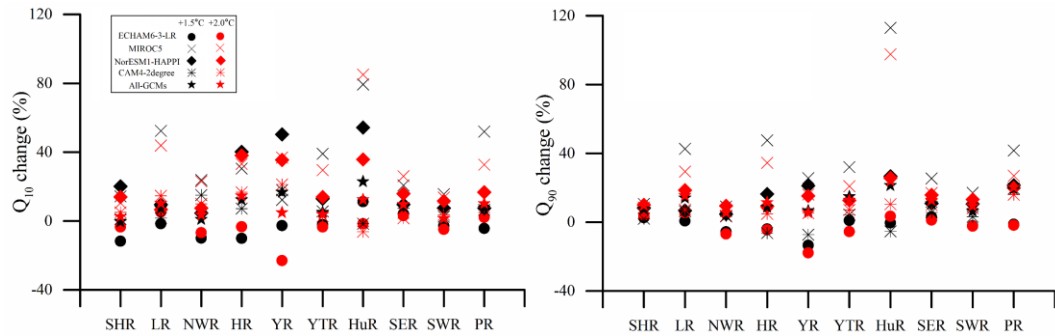

**Figure A5: Changes (%) in low runoff ($Q_{10}$) (a) and high runoff ($Q_{90}$) (b) in each main basin in China in all corresponding ensembles of the four GCMs under 1.5°C and 2.0°C warming scenarios in 2106-2115, respectively, relative to 2006-2015.**

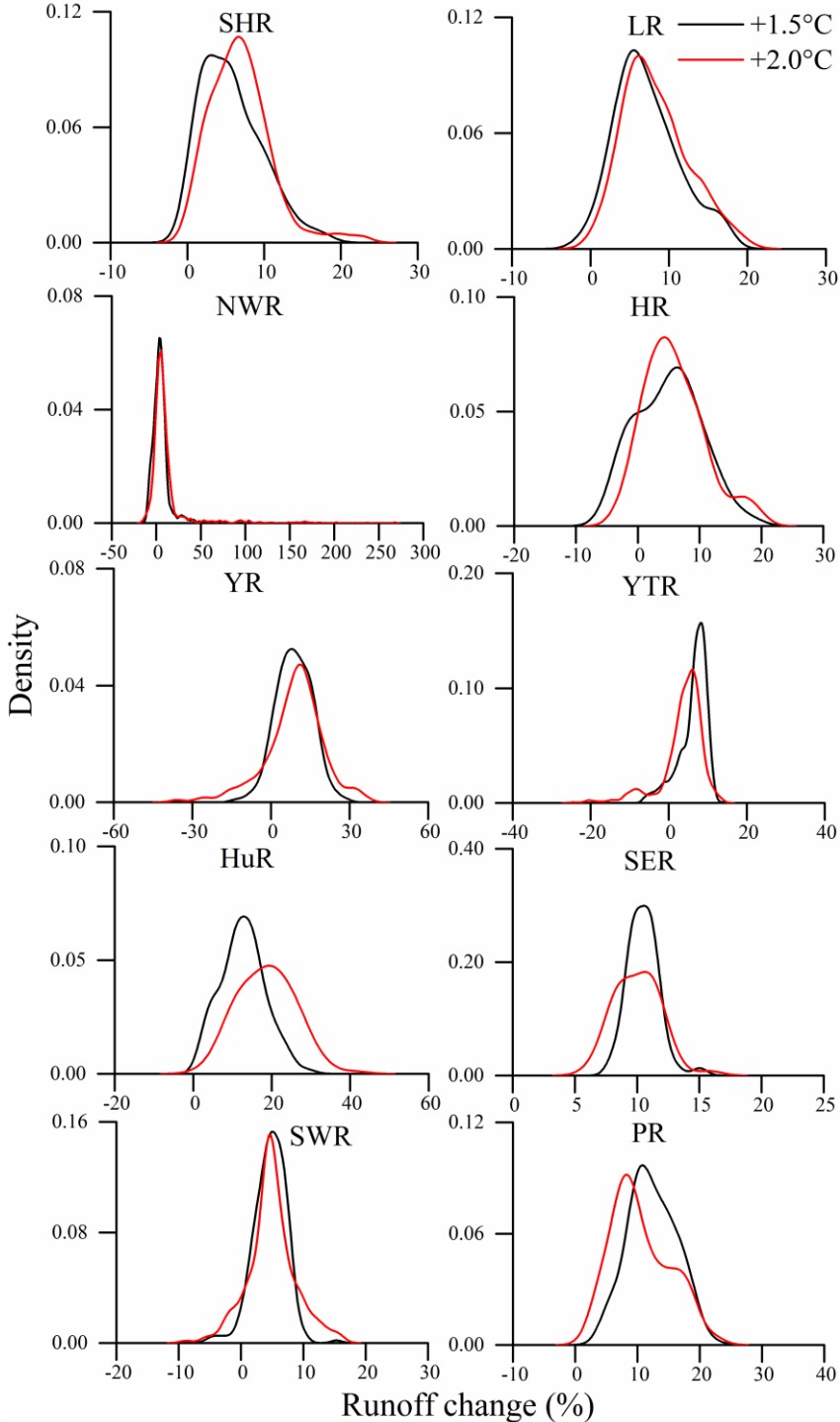

**Figure A6: Probability density functions of runoff change (%) under 1.5°C and 2.0°C warming scenarios in each basin, relative to 2006-2015.**

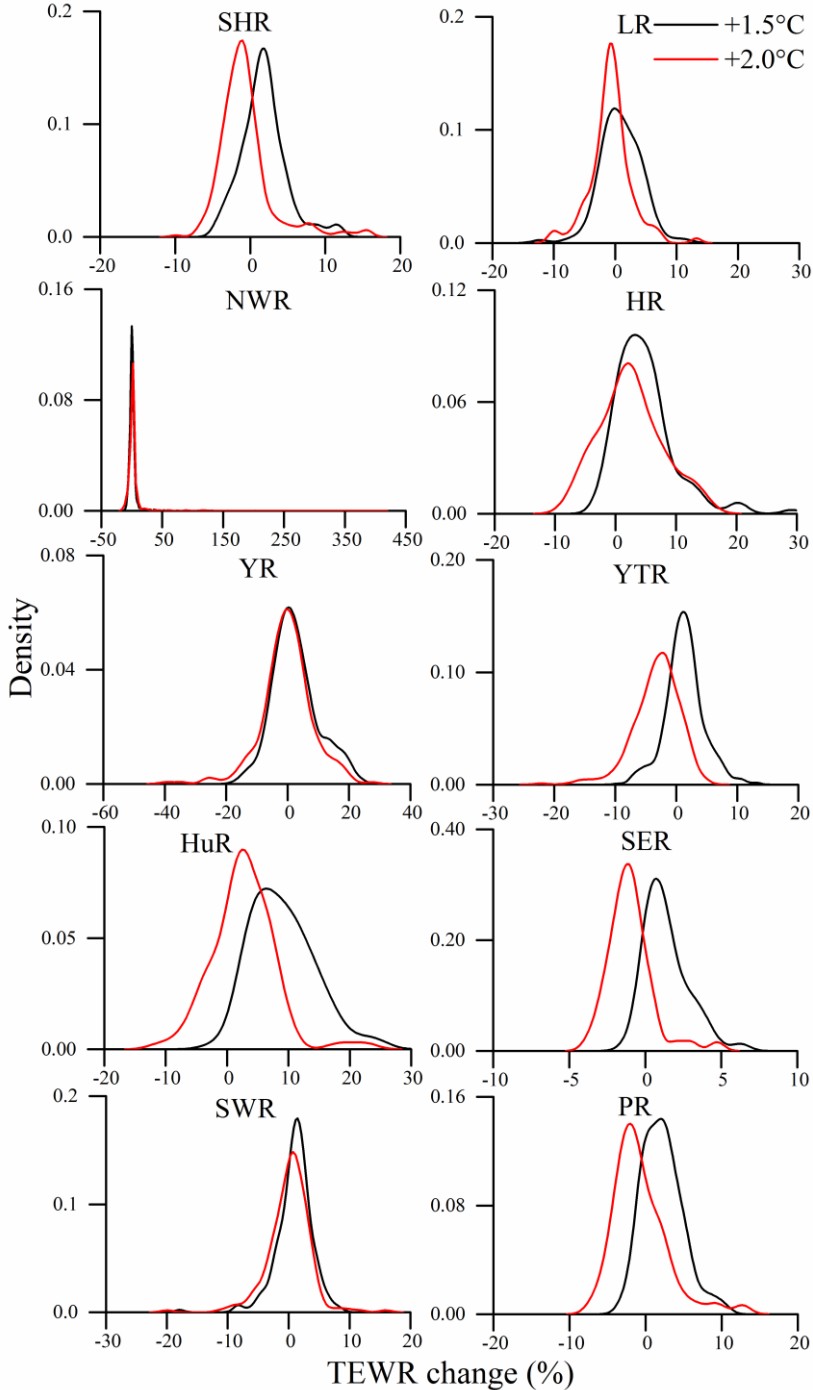

**Figure A7: Probability density functions of TEWR change (%) under 1.5°C and 2.0°C warming scenarios in each basin, relative to 2006-2015.**

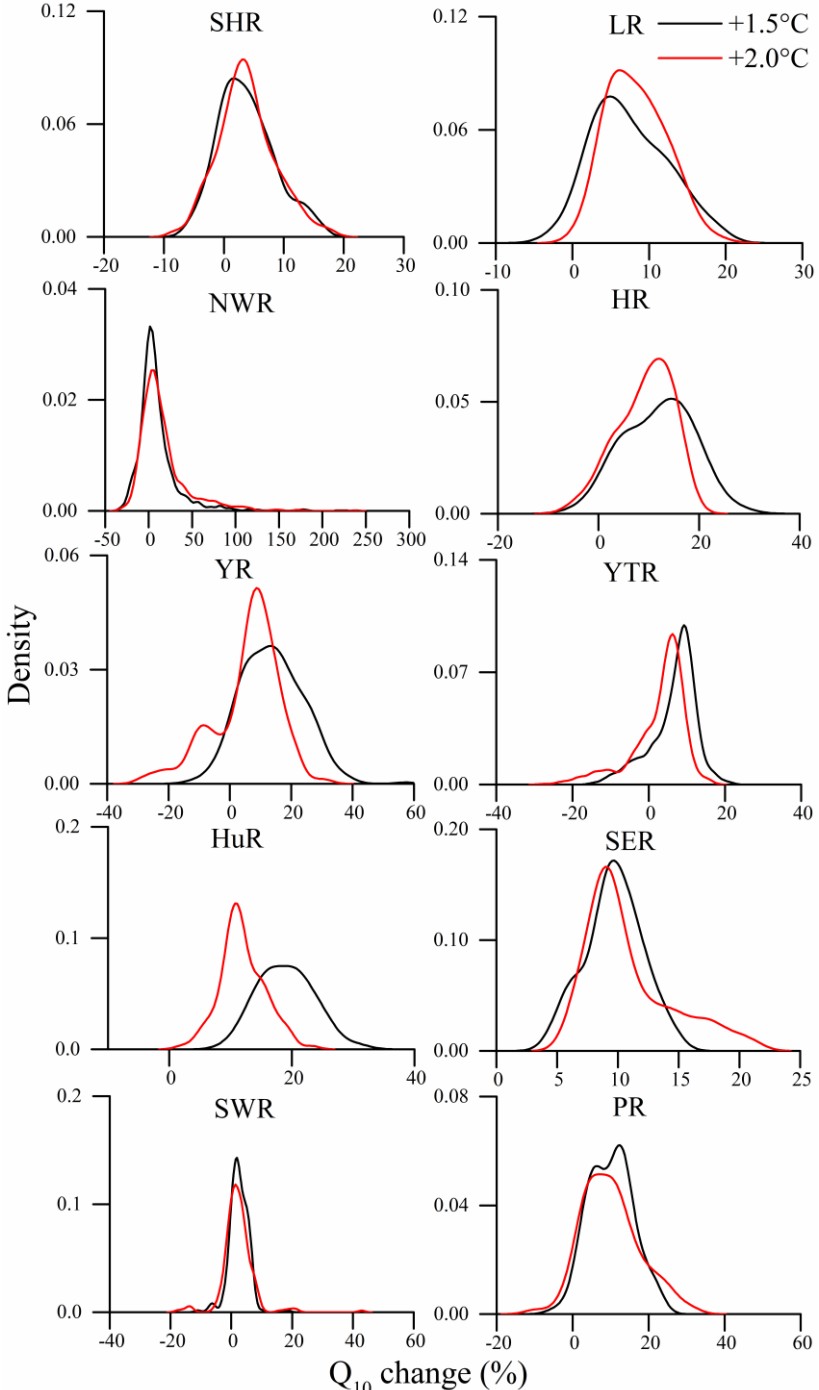

**Figure A8: Probability density functions of low runoff change (%) under 1.5°C and 2.0°C warming scenarios in each basin, relative to 2006-2015.**

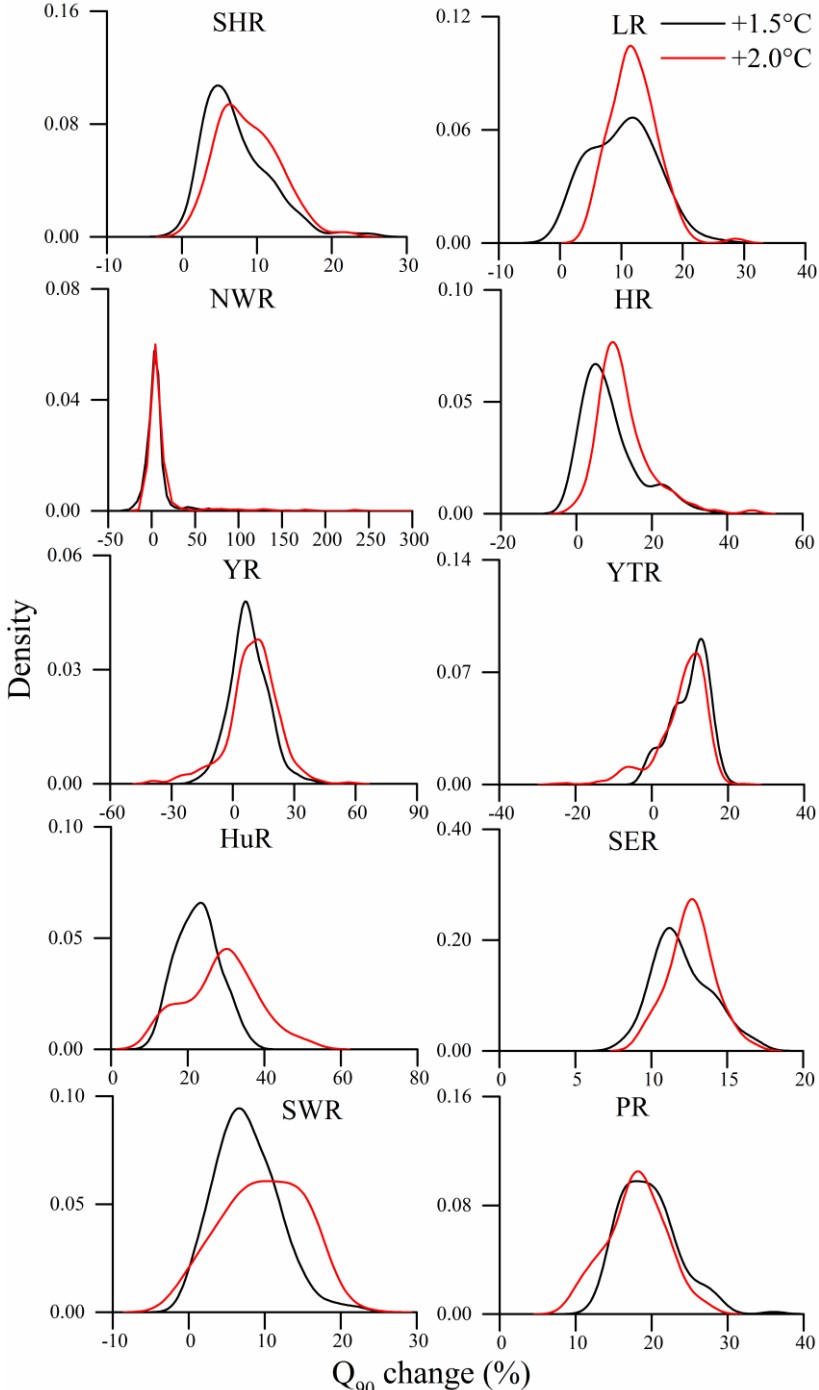

**Figure A9: Probability density functions of high runoff change (%) under 1.5°C and 2.0°C warming scenarios in each basin, relative to 2006-2015.**