# Peer review of "Spatial-temporal changes in runoff and terrestrial ecosystem water retention under 1.5°C and 2°C warming scenarios across China"

_Earth System Dynamics, 2017_

## Referee Comment (RC1) · Anonymous Referee #1 · 11 Dec 2017

The manuscript investigated the impacts of global warming on river runoff and terrestrial ecosystem water retention under two climate scenarios. Overall, I feel the paper is well-written and presents interesting results. However, some detailed explanations and more analysis are needed. Finally, I'd recommend the paper for publication after substantial improvements have been made to address the following concerns (major revision).

Comments: 1. The authors should present the full name of some abbreviations for the first occurrence. For example, "GCMs", "ECHAM6-3-LR, MIROC5, NorESM1-HAPPI, and CAM4-2degree" (Pages 2 Line 30, Page 3 Line 1). 2. Page 4 Line: "showed"

[Figure]

should be "shows". 3. Do Figs. 2 and 3 show the changes in temperature and precipitation under the two warming scenarios relative to the baseline period (2006-2015)? In addition, "all the four GCMs" in Figs. 2e and 2j is the average mean of the four GCMs? Please make them clear. 4. Page 6 Line 20: "Fig. 2a, d, f, i" should be "Figs. 2a, d, f, i". The same for the whole manuscript. 5. Compared to the mean value, the projections of hydrological extremes are more important. However, the explaining for the reason of the changes in Q10 and Q90 and the differences between 1.5 and 2.0 warming scenarios were not presented. 6. Fig. 10 showed the Pearson correlations between river runoff and maximum and minimum temperatures, since the last two climate variables were the input of the VIC model. However, the manuscript mainly discussed the mean temperature in Section 3. In addition, wind speed was not mentioned in Sections 2 and 3. 7. The Spearman correlation coefficients did not represent the contributions of the input climate variables. The authors should use other statistical methods, such as partial correlations or multi-regression method, to investigate the contribution of each climate variable on the river runoff and TEWR. 8. The description in Section 4.3 did not provide any useful information for the readers. It is better to evaluate the performances of the VIC model driven by GCM ensemble simulations using ground-based or satellite-based observations between 2006 and 2015.

---

## Referee Comment (RC2) · Anonymous Referee #2 · 3 Jan 2018

This paper presented changes in river runoff and terrestrial ecosystem water retention in China under 1.5 and 2 degree warming scenarios simulated by the VIC model. The simulations are under the HAPPI project framework, and the results are interesting in the context of Paris Agreement and IPCC 1.5 degree special report. There are however major issues in the current version of the manuscript that should be addressed before considering the paper for publication.

1. Although previous research has not investigated the changes in river runoff under 1.5 and 2 degree warming specifically, it is useful to compare with changes under RCP2.6 scenario (the warming level is similar, and aerosol forcing used is also the

same as RCP2.6's end of 21st century level).

2. For the wide readership of ESD, it would be helpful to expand the model description of VIC and explain briefly the key model features. More details should also be provided for the calibration procedure and validation. Ideally different stations could be used for the validation purpose. It would also be interesting if the authors would briefly explain why the bias for the calibration and validation period is of opposite direction for all stations (Table 2).

3. The uncertainty analysis is currently lacking. Section 4.3 proposed a few potential uncertainty sources but no actual analyses is presented. A better effort to present uncertainty in the results would improve the readability of the paper. For example, for some of the projected changes the authors could show maps of model agreement (number of model ensembles showing the same trend) at each grid point. It is not clear if the changes are statistically significant. The uncertainty range among the ensembles for the GCMs should also at least be mentioned for Figure 9.

4. A more quantatitive conclusion should be provided with some regionally averaged statistics, especially on the difference between 1.5 degree and 2 degree scenarios. In general more quantatitive results should be provided from the analyses as well. The connection to climate change mitigation and adaptation needs is not convincing as the manuscript discussed very little about the risks (which should also have a socioeconomics component) of hydrological extremes, and the effect of human management is also not examined in this study. The last paragraph is a bit far-fetched and is not supported well enough by the results.

A couple of minor points/questions:

- Are there similar results from HAPPI based on a different model other than VIC?

- What is the role of snow accumulation/melting change, especially in/near the Tibetan Plateau?

- What is the reasoning of showing change in Q10 and Q90 based on the full ensemble, instead of for each GCM (similar to the other figures)? Will the results vary among the GCMs? It is also arguable how "extreme" Q10 and Q90 are.

- Figure 5 seems to have a more mixed result on the change of river runoff SD: in some regions the SD increase is larger under 1.5 degree scenario especially for NorESM1.

- How large is the signal of half degree warming compared to model specific difference in precipitation pattern? Figure 8 for example show quite different spatial patterns of changes among the GCMs (the first sentence on page 9 is a bit confusing, please reword).

- Both runoff (generated at grid level) and river runoff are used which may confuse the reader. Is river runoff river discharge (after routing)?

- Table 2 should include units for catchment area, and explain how model grid is selected (note for Zhongaiqiao station, its longitude is at the edge of a 0.5 degree grid box).

---

## Author Comment (AC1) · 8 Feb 2018

Thanks a lot for the positive comments. We have taken all of them seriously and had major revisions accordingly. Our responses by point-to-point are detailed in blue, and the revised portions are marked in red in the revised version. The Response to Reviewer 1 and the revised manuscript were all in the supplement.

Please also note the supplement to this comment:
https://www.earth-syst-dynam-discuss.net/esd-2017-96/esd-2017-96-AC1-supplement.pdf

---

## Author Response (AR1)

**Dear Editor,**

Thank you and the two reviewers so much for the insightful comments and suggestions. We have taken all of them seriously and had major revisions accordingly. Our responses by point-to-point are detailed below in blue, and the revised portions are marked in red in the revised version. Your reconsiderations will be highly appreciated!

Sincerely yours

Fulu Tao & Ran Zhai

**Response to Reviewer #1:**

The manuscript investigated the impacts of global warming on river runoff and terrestrial ecosystem water retention under two climate scenarios. Overall, I feel the paper is well-written and presents interesting results. However, some detailed explanations and more analysis are needed. Finally, I'd recommend the paper for publication after substantial improvements have been made to address the following concerns (major revision).

Thanks a lot for the positive comments.

1. The authors should present the full name of some abbreviations for the first occurrence. For example, "GCMs", "ECHAM6-3-LR, MIROC5, NorESM1-HAPPI, and CAM4-2degree" (Pages 2 Line 30, Page 3 Line 1).

Revised. We have presented the full name of all abbreviations for the first occurrence (Page 2 line 9, Page 2 lines 16-17, Page 2 line 19, Page 2 line 27, Page 4 line 3). And we added original institute name and original institute ID of ECHAM6-3-LR, MIROC5, NorESM1-HAPPI and CAM4-2degree model in Table 1.

2. Page 4 Line: "showed" should be "shows".

We have revised it in Page 4 line 8.

3. Do Figs. 2 and 3 show the changes in temperature and precipitation under the two warming scenarios relative to the baseline period (2006-2015)? In addition, "all the four GCMs" in Figs. 2e and 2j is the average mean of the four GCMs? Please make them clear.

Yes, Figs. 2 and 3 show the changes in temperature and precipitation under the two warming scenarios relative to the baseline period (2006-2015), it was explained in the revised manuscript in Section 2.6 (Page 5 lines 20-22). The captions of Figs. 2 and 3 were revised to clarify these. "all the four GCMs" in Figs. 2e and 2j are the median values of the projected changes among

all 70 ensembles under the four GCMs (Page 5 line 28 to Page 6 line 1).

4. Page 6 Line 20: "Fig. 2a, d, f, i" should be "Figs. 2a, d, f, i".The same for the whole manuscript.

We have revised all these in the manuscript.

5. Compared to the mean value, the projections of hydrological extremes are more important. However, the explaining for the reason of the changes in Q10 and Q90 and the differences between 1.5 and 2.0 warming scenarios were not presented.

We presented the differences of $Q_{10}$ and $Q_{90}$ between 1.5°C and 2.0°C warming scenarios in Section 3.4, and added a Figure about SD of precipitation (Fig. A3) to explain for the reason of the changes in $Q_{10}$ and $Q_{90}$ (Page 8, lines 23-24). And we added Figure A8 and Figure A9 to show the probability density functions of changes of $Q_{10}$ and $Q_{90}$ under 1.5°C warming scenario and 2.0°C warming scenario across all 700 years in 70 ensembles in the four GCMs for all the grids in each basin. Results showed most grids under the two warming scenarios showed increasing $Q_{10}$ and $Q_{90}$ across China. $Q_{90}$ was projected to increase more in most basins across China than $Q_{10}$ under 2.0°C warming scenario than 1.5°C warming scenario (Figs. A8 and A9). And this may imply more flood and drought risks under 2.0°C warming scenario than 1.5°C warming scenario. Please refer to Page 10 line 29 to Page 11 line 2.

6. Fig. 10 showed the Pearson correlations between river runoff and maximum and minimum temperatures, since the last two climate variables were the input of the VIC model. However, the manuscript mainly discussed the mean temperature in Section 3. In addition, wind speed was not mentioned in Sections 2 and 3.

Yes, we only discussed annual mean temperature and annual precipitation in Section 3, because these two climate variables are the most representative and important variables to represent every scenario of each GCM's characteristics. Only key variables and results were stressed to avoid too many figures. And we added this point in Section 2.3 in Page 4 lines 9-11.

7. The Spearman correlation coefficients did not represent the contributions of the input climate variables. The authors should use other statistical methods, such as partial correlations or multi-regression method, to investigate the contribution of each climate variable on the river runoff and TEWR.

We tried to use multi-regression method and partial correlation method to investigate the contribution of each variable on runoff and TEWR. But the results were strange. For example, the relation between annual runoff change and annual maximum temperature change after standardization through multi-regression method was shown in Fig. R1. According to common sense,

maximum temperature would lead to runoff decrease because of increasing evapotranspiration in most areas in China (Liu et al., 2017), but the results there showed annual maximum temperature change had a positive relation with runoff change in most areas across China. We think it was caused by multicollinearity between annual maximum temperature and annual minimum temperature. For partial correlation method, we need to control other independent variables and analyze the relation between one independent variable and dependent variable. It is unrealistic to only change maximum temperature without change the minimum temperature in the real world. So, we only analyzed the correlation between runoff and the four input variables (precipitation, maximum temperature, minimum temperature, wind speed). As for TEWR, because we used a simple linear equation to calculate TEWR, so the coefficients were consistent with the equation when using multi-regression method and partial correlation method. So, we just used the Pearson correlation method to show the relationships between the independent variables and the dependent variables, although the Pearson correlation coefficient did not represent the contribution of the input climate variables.

[Figure]

**Figure R1: Spatial patterns of the multi-regression coefficient (r) between data series of runoff changes and four key impact factors**

**changes (a: runoff and precipitation, b: runoff and annual maximum temperature, c: runoff and annual minimum temperature, d: runoff and wind speed) under 1.5 °C and 2.0 °C warming scenarios from 2106 to 2115 relative to the baseline period from 2006 to 2015.**

8. The description in Section 4.3 did not provide any useful information for the readers. It is better to evaluate the performances of the VIC model driven by GCM ensemble simulations using ground-based or satellite-based observations between 2006 and 2015.

Thanks a lot for the insightful suggestions. We added three parts to make our uncertainties more meaningful. First, according to suggestion, we evaluated the performances of the VIC model driven by GCM ensembles simulations using ground-based observation climate data between 2006 to 2015. We analyzed the differences between the median value of VIC model result driven by 70 GCM ensembles simulations and driven by ground-based observation data between 2006 to 2015. Please refer to Page 12 lines 20-30, and Fig 12. Second, to make our results more convincing, we added other ten observation runoff station data located in different main basins to validate our calibrated parameters, and the result showed our calibrated parameters were also validated for other catchments in the same basin. Please refer to Page 6 line 26 to Page 7 line 3 and Table 3. Third, we evaluated the model agreement through analyzing the number of model ensembles showing the same increasing trend in runoff and TEWR at each grid, please refer to Page 12 line 31 to Page 13 line 2 and Figure 13. Based on all of these, we think Section 4.3 now provide more useful information for readers than before.

Reference

Liu, J. Y., Zhang, Q., Singh, V. P., and Shi, P. J.: Contribution of multiple climatic variables and human activities to streamflow changes across China, Journal of Hydrology, 545, 145-162, 10.1016/j.hydrol.2016.12.016, 2017.

**Response to Reviewer #2:**

This paper presented changes in river runoff and terrestrial ecosystem water retention in China under 1.5 and 2 degree warming scenarios simulated by the VIC model. The simulations are under the HAPPI project framework, and the results are interesting in the context of Paris Agreement and IPCC 1.5 degree special report. There are however major issues in the current version

of the manuscript that should be addressed before considering the paper for publication.

Thanks a lot for the positive comments.

1. Although previous research has not investigated the changes in river runoff under 1.5 and 2 degree warming specifically, it is useful to compare with changes under RCP2.6 scenario (the warming level is similar, and aerosol forcing used is also the same as RCP2.6's end of 21st century level).

Revised. We compared the results with other studies under RCP2.6 scenario across China. Please refer to the revised manuscript in Page 12 lines 10-19.

2. For the wide readership of ESD, it would be helpful to expand the model description of VIC and explain briefly the key model features. More details should also be provided for the calibration procedure and validation. Ideally different stations could be used for the validation purpose. It would also be interesting if the authors would briefly explain why the bias for the calibration and validation period is of opposite direction for all stations (Table 2).

Firstly, following the suggestions, we expanded the model description of VIC in Section 2.2 (Page 3 lines 20-27). Secondly, to provide more details for the calibration and validation procedure in Section 3.1, we added Fig. A1, and the parameters needed to be calibrated in Page 4 lines 25-28. Thirdly, we collected ten other hydrological stations located in different main basins for the validation purpose in Section 3.1 (Page 6 line 26 to Page 7 line 3), and we added Table 3 to show these hydrological stations characteristics and validation results to support our results. We also added the locations of these ten stations in Figure 1. Lastly, as for the bias for the calibration and validation procedure, it might be caused by trying to make the results of calibration and validation both better when parameters were calibrated.

3. The uncertainty analysis is currently lacking. Section 4.3 proposed a few potential uncertainty sources but no actual analyses is presented. A better effort to present uncertainty in the results would improve the readability of the paper. For example, for some of the projected changes the authors could show maps of model agreement (number of model ensembles showing the same trend) at each grid point. It is not clear if the changes are statistically significant. The uncertainty range among the ensembles for the GCMs should also at least be mentioned for Figure 9.

Thanks a lot for the insightful suggestions. We added three parts to make our uncertainties more meaningful. First, we evaluated the model agreement through analyzing the number of model ensembles showing the same increasing trend in runoff

and TEWR at each grid, please refer to Page 12 line 31 to Page 13 line 2 (Fig. 13). Second, we added Figure A4 to show the uncertainty range for annual mean temperature change, annual precipitation change, annual runoff change and annual TEWR change among the ensembles for all the GCMs. Third, we evaluated the performances of the VIC model driven by GCM ensembles simulations using ground-based observation climate data between 2006 to 2015. We analyzed the differences between the median value of VIC model result driven by 70 GCM ensembles simulations and driven by ground-based observation data between 2006 to 2015. Please refer to Page 12 lines 20-30 (Fig 12). Based on all of these, we think Section 4.3 now provide more useful information for readers than before.

4. A more quantatitive conclusion should be provided with some regionally averaged statistics, especially on the difference between 1.5 degree and 2 degree scenarios. In general more quantatitive results should be provided from the analyses as well. The connection to climate change mitigation and adaptation needs is not convincing as the manuscript discussed very little about the risks (which should also have a socioeconomics component) of hydrological extremes, and the effect of human management is also not examined in this study. The last paragraph is a bit far-fetched and is not supported well enough by the results.

Firstly, we added Table 4 to provide with some regional averaged statistics data, to show how annual temperature, annual precipitation, annual runoff and annual TEWR change under 1.5℃ and 2.0℃ warming scenarios, and to show the difference between 1.5℃ and 2.0℃ warming scenarios under all ensembles in the four GCMs. And we also added some quantitative results in Section 4.1, please refer to Page 9 lines 26-29, Page 9 line 30 to Page 10 line 1, Page 10 lines 7-10 and Page 10 lines 14-15. Secondly, following the suggestion, we deleted the last paragraph in conclusion.

A couple of minor points/questions:

1. Are there similar results from HAPPI based on a different model other than VIC?

Based on our knowledge, there are only a few studies using HAPPI data now, however, this is the first to use HAPPI data to analyze the runoff and TEWR change in China.

2. What is the role of snow accumulation/melting change, especially in/near the Tibetan Plateau?

As previous studies, snowmelt contributes much water to the runoff for rivers in the Tibetan Plateau (Liu et al., 2015). However, snow accumulation/melting change is a complicated process (Su et al., 2016). Under warming scenarios, first snow melting

would lead to runoff increase (Zhang et al., 2013). And the time of snow melting would in advance and change the annual runoff distribution (Barnett et al., 2005). However, because of global warming, glacial retreat and the amount of snow accumulation decrease, the runoff caused by snow and glacier melting would decrease later (Liu et al., 2015). And in this present study, unlike other main basins, runoff had a weaker relation with precipitation in some area in the Tibetan Plateau (Fig. 10a), and increasing annual minimum temperature would lead to runoff increase (Fig. 10c), which may cause by snow and glacier melting in the Tibetan Plateau. We added this point in Page 11, lines 20-23.

3. What is the reasoning of showing change in Q10 and Q90 based on the full ensemble, instead of for each GCM (similar to the other figures)? Will the results vary among the GCMs? It is also arguable how "extreme" Q10 and Q90 are

It is more meaningful to use large number of sample data to calculate $Q_{10}$ and $Q_{90}$, so we used all the ensemble data instead of every ensemble data in each GCM. In the revised manuscript, we added a figure to illustrate the $Q_{10}$ and $Q_{90}$ change in each GCM in every main basin (Fig. A5). It was true that the results of the changes in $Q_{10}$ and $Q_{90}$ in each basin were different for different GCM, so using a large sample data to analyze would be more comprehensive and convincing to present a general idea of the changes in $Q_{10}$ and $Q_{90}$.

As for $Q_{10}$ and $Q_{90}$, $Q_{10}$ represents ten percentile value in all the 700 years data, and $Q_{90}$ represent ninety percentile value in all the 700 years data in each grid in each period (2006-2015, 2106-2115 under 1.5°C warming scenario, and 2106-2115 under 2.0°C warming scenario). They could represent high runoff and low runoff change under the two warming scenarios. Although, there are some other indicators, for example $Q_{15}$ (low runoff) and $Q_{85}$ (high runoff), $Q_5$ (low runoff) and $Q_{95}$ (high runoff), but $Q_{10}$ and $Q_{90}$ were used more to represent extreme events (Strzepek et al., 2013;Krysanova et al., 2017;Vetter et al., 2017). If $Q_{10}$ decrease more, it may represent more drought risks, and if $Q_{90}$ increase more, it may represent more flood risks in a period. Because $Q_{10}$ and $Q_{90}$ of yearly runoff may not represent extreme droughts and floods only happened in several days, so we deleted 'extreme' in our revised manuscript, and only used low runoff and high runoff to describe $Q_{10}$ and $Q_{90}$.

4. Figure 5 seems to have a more mixed result on the change of river runoff SD: in some regions the SD increase is larger under 1.5 degree scenario especially for NorESM1.

It is true that the SD increase is larger under 1.5°C warming scenario in some regions. Different GCMs may give contrasting results, but in general, the increase of SD was larger under 2.0°C warming scenario than 1.5°C warming scenario. Because of

the large area of our study region, it is hard to get a uniform result in every region and in every GCM. So we analyzed median values of all ensembles in the four GCMs to draw a more general, comprehensive and convincing result (Page 13 lines 25-26).

5.    How large is the signal of half degree warming compared to model specific difference in precipitation pattern? Figure 8 for example show quite different spatial patterns of changes among the GCMs (the first sentence on page 9 is a bit confusing, please reword).

First, according to Figure 9b, we found model specific difference in precipitation pattern was much larger than half degree warming, because differences between precipitation change under different scenarios of the same GCM were smaller than differences between precipitation change under the same scenario of different GCMs. And this finding was supported by many other researches (Chen et al., 2011; Ouyang et al., 2015; Liu et al., 2017; Zhang et al., 2017). For example, Ouyang et al. (2015) found that GCM models and RCPs would suggest large uncertainty in the future climate scenarios, and GCM modeling was a large source of uncertainty than variations in RCPs. So, we tried to get a more comprehensive result through using 4 GCMs including 70 ensembles. And we added this point in our manuscript in Page 10 lines 10-12 and Page 13 lines 25-26. Second, we rewrote the first sentence on Page 9, please refer to Page 9, lines from 16 to 17 now.

6.    Both runoff (generated at grid level) and river runoff are used which may confuse the reader. Is river runoff river discharge (after routing)?

No, river runoff also means runoff generated at grid scale. To avoid confusion, we changed all river runoff to runoff in the manuscript.

7.    Table 2 should include units for catchment area, and explain how model grid is selected (note for Zhongaiqiao station, its longitude is at the edge of a 0.5 degree grid box).

Yes, we have added units ($km^2$) for catchment area in Table 2. And explained how model grid was selected (Page 3 lines 18-20). And Zhongaiqiao station is located in the edge of two grids, it only had effect when we routed runoff to calibrate and validate the model using observation data, we need to confirm which grid contained the station, so we chose the grid which had a lower elevation according to elevation data, because water always flow downwards.

[revised manuscript text omitted]

---

## Author Response (AR2)

**Dear Editor,**

Thank you and the reviewer so much for the insightful comments and suggestions. We have taken all of them seriously and had a minor revision accordingly. Our responses by point-to-point are detailed below in blue, and the revised portions are marked in red in the revised version. Your reconsideration will be highly appreciated!

Sincerely yours

Fulu Tao & Ran Zhai

**Response to Reviewer #2:**

The paper has significantly improved in revision. I only have some minor comments which should be easy to address.

Thanks a lot for the positive comments.

1. In the model description it is important to also describe the evaporation scheme and runoff scheme used.

Revised. We added the evapotranspiration scheme and runoff scheme of the VIC model, and added a reference. Please refer to Page 3 line 24 to Page 4 line 2.

2. It is not clear what the authors mean by "for the calibration and validation procedure, it might be caused by trying to make the results of calibration and validation both better when parameters were calibrated". The validation should be independent of calibration. Is this bias possibly related to flow magnitude? It would be nice to show mean annual flow magnitude for the stations in the tables.

Yes, the bias is related to flow magnitude. The differences of the annual mean runoff between calibration period and validation period were caused by complicated reasons, such as climate change, human activities although there were less influence before 1980 than after 1980 in China, and the peak value as well, VIC model could not simulate the peak value accurately, noticing the Qinan station, the *BIAS* was negative in the validation period because of the really high peak value (Fig. S1). And if there existed increased water withdrawal at a catchment, the observed annual mean runoff may decrease in the validation period than the calibration period. So the projected runoff may be high in the validation period using the parameters calibrated in the previous time. When we validated the calibrated parameters in the validation time, if the /*BIAS*/ was large, or the *NSE* was small, we would go back and calibrate the parameters again. And our calibrated parameters were also validated in ten different catchments, which made our results more convincing. We added the annual mean runoff for the first ten stations used for calibrating and validating the parameters, and the second other ten stations only used for validating the parameters. Please refer to Table 2 and Table 3.

3. In Figure A4 it would be nice to show symbols of individual models on the box plot.

Revised. Please refer to the new Figure S4 now.

4. The new quantitative results and comparison with earlier studies using RCP2.6 is very good. However the abstract and conclusion in the current form are still mostly qualitative. I would like to ask the authors regarding the key message of this paper: is the consistency of projected change or the large uncertainty more important? A revisit on the abstract and conclusion maybe necessary depending on the answer.

We added some quantitative results in the abstract and conclusion. Please refers to lines 18-20 and lines 22-23 in Page 1, line 30 in Page 13 to line 4 in Page 14, lines 11-14 in Page 14. In this paper, the consistency of projected change is more important. We want to show the general changing trends in annual mean temperature, annual precipitation, annual runoff and annual TEWR in China although there exist large uncertainties among GCMs. This study applied ensemble projections from multiple GCMs to provide more comprehensive and robust results. And our results were supported by previous studies. According to the answer, we revised the abstract and conclusion.

5. Sorry I missed this earlier: I know there are some inconsistency in the literature, but Q10 usually refers to high flow (magnitude of runoff that is exceeded 10% of the time). Therefore I suggest the authors to switch the terms. If the authors instead would like to use it for low flow, please give a clear definition.

Revised. Thank you for pointing out our mistakes. $Q_{10}$ refers to high flow and $Q_{90}$ refers to low flow in the revised manuscript. Please refer to lines 16-17 in Page 6.

[revised manuscript text omitted]